# Improving Differentially Private SGD via Randomly Sparsified Gradients

**Junyi Zhu**                                                               *junyi.zhu@esat.kuleuven.be*
*Center for Processing Speech and Images, Department of Electrical Engineering (ESAT)*
*KU Leuven, Belgium*

**Matthew B. Blaschko**                                              *matthew.blaschko@esat.kuleuven.be*
*Center for Processing Speech and Images, Department of Electrical Engineering (ESAT)*
*KU Leuven, Belgium*

**Reviewed on OpenReview:** *https://openreview.net/forum?id=sY35BAiIf4*

## Abstract

Differentially private stochastic gradient descent (DP-SGD) has been widely adopted in deep learning to provide rigorously defined privacy, which requires gradient clipping to bound the maximum norm of individual gradients and additive isotropic Gaussian noise. With analysis of the convergence rate of DP-SGD in a non-convex setting, we identify that randomly sparsifying gradients before clipping and noisification adjusts a trade-off between internal components of the convergence bound and leads to a smaller upper bound when the noise is dominant. Additionally, our theoretical analysis and empirical evaluations show that the trade-off is not trivial but possibly a unique property of DP-SGD, as either canceling noisification or gradient clipping eliminates the trade-off in the bound. This observation is indicative, as it implies DP-SGD has special inherent room for (even simply random) gradient compression. To verify the observation an utilize it, we propose an efficient and lightweight extension using random sparsification (RS) to strengthen DP-SGD. Experiments with various DP-SGD frameworks show that RS can improve performance. Additionally, the produced sparse gradients of RS exhibit advantages in reducing communication cost and strengthening privacy against reconstruction attacks, which are also key problems in private machine learning.

## 1 Introduction

Internet-scale data promises to accelerate the development of data-driven statistical approaches, but the need for privacy constrains the amalgamation of such datasets. As a result, private data are in fact isolated, constraining our ability to build models that learn from a large number of instances. On the other hand, the information contained in locally stored data can also be exposed through releasing the model trained on a local dataset (Fredrikson et al., 2015; Shokri et al., 2017), or even reconstructed when gradients generated during training are shared (Zhu et al., 2019; Zhu & Blaschko, 2021; Zhu et al., 2023).

To address these issues, many applications of machine learning are expected to be privacy-preserving, while differential privacy (DP) provides a rigorously defined and measurable privacy guarantee. As described in Definition 1, DP defines privacy with respect to the difficulty of distinguishing the outputs of different data:

**Definition 1** (($\varepsilon, \delta$)-DP (Dwork & Roth, 2014))**.** For a pair of neighboring datasets $X, X' \in \mathcal{X}$, $X$ is obtained from $X'$ by adding or removing an element. A randomized mechanism $\mathcal{M} : \mathcal{X} \to \mathcal{R}$ is ($\varepsilon, \delta$)-differentially private, if for any subset of outputs $S \subseteq \mathcal{R}$ it holds that:

$$\Pr[\mathcal{M}(X) \in S] \le e^\varepsilon \Pr[\mathcal{M}(X') \in S] + \delta. \tag{1}$$

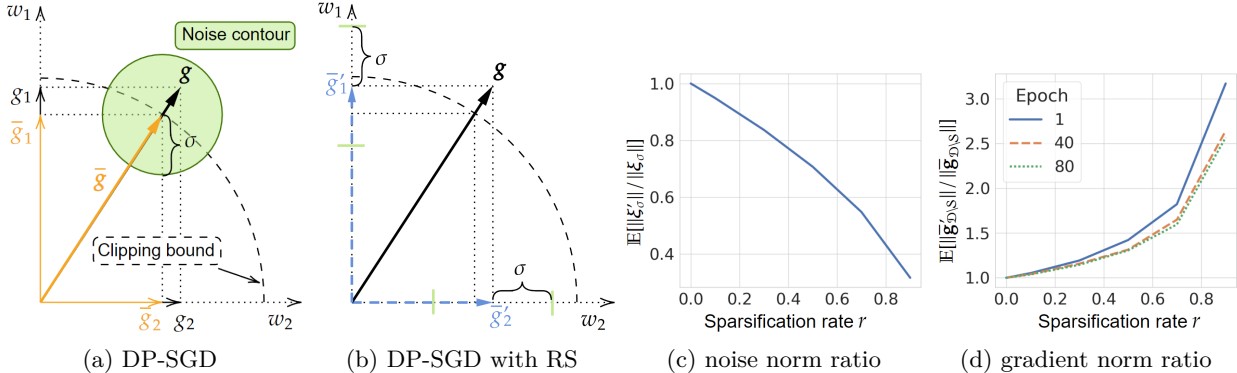

Figure 1: **(a)** Schematic of DP-SGD. We denote $\boldsymbol{g}$ as the gradient vector and omit the notation for input data, $g_1$ and $g_2$ are two entries in $\boldsymbol{g}$ corresponding to the coordinates $w_1$ and $w_2$, $\bar{\boldsymbol{g}}$ is the clipped gradient vector. **(b)** Schematic of DP-SGD with RS, $\bar{g}'_1$ and $\bar{g}'_2$ are two possible results under RS. **(c)** Empirical ratio of noise norm with RS $\|\boldsymbol{\xi}'_\sigma\|$ and without RS $\|\boldsymbol{\xi}_\sigma\|$, i.e. $\|\boldsymbol{\xi}'_\sigma\|/\|\boldsymbol{\xi}_\sigma\|$, we take $d = 5.5 \times 10^5$ which is the dimension of DP-CNN. **(d)** Empirical ratio of the gradient norm of relaxed coordinates with RS $\|\bar{\boldsymbol{g}}'_{\mathcal{D}\backslash\mathcal{S}}\|$ and without RS $\|\bar{\boldsymbol{g}}_{\mathcal{D}\backslash\mathcal{S}}\|$, i.e. $\|\bar{\boldsymbol{g}}'_{\mathcal{D}\backslash\mathcal{S}}\|/\|\bar{\boldsymbol{g}}_{\mathcal{D}\backslash\mathcal{S}}\|$, measured on DP-CNN with dataset CIFAR10. We remark that for (a) and (b) clipping bound $C$ within range $(\max(\{|g_i|\}_{i=1}^d), \|\boldsymbol{g}\|)$ is representative, as in high-dimensional space $\|\boldsymbol{g}\| \gg \max(\{|g_i|\}_{i=1}^d)$, while the two strengths still exist for $C$ being smaller.

A common paradigm for applying DP in deep learning is differentially private stochastic gradient descent (DP-SGD) proposed by Abadi et al. (2016b), which lets the randomized mechanism $\mathcal{M}$ output perturbed gradients:

$$f(X) := \sum_{x \in X} g(x), \tag{2}$$

$$\mathcal{M}(X) := f(X) + \mathcal{N}(0, S_f^2 \sigma^2 \boldsymbol{I}_d), \tag{3}$$

where $X$ now stands for a batch of data and $g(x) \to \mathbb{R}^d$ computes gradient given an individual example in the batch. The isotropic Gaussian distributed noise $\mathcal{N}(0, S_f^2 \sigma^2 \boldsymbol{I})$ is calibrated to $f$'s sensitivity $S_f^2 := \max_{X,X'} \|f(X) - f(X')\|$, while the noise multiplier $\sigma$ controls the strength of the privacy guarantee. As $S_f$ is usually unknown, DP-SGD commonly clips the gradient of each individual example in Euclidean norm to a preset bound C, such that $\bar{f}(X) := \sum_{x \in X} g(x) \cdot \min(1, C/\|g(x)\|)$. Then $S_f$ can be safely replaced by $C$ and the randomized mechanism $\mathcal{M}$ can thus be expressed as:

$$\mathcal{M}(X) := \bar{f}(X) + \mathcal{N}(0, C^2 \sigma^2 \boldsymbol{I}_d). \tag{4}$$

However, gradient clipping gives rise to a squeezed gradient distribution, while noisification further blurs the gradient distribution, both cause adverse effects for optimization. Understanding the influence of these two operations becomes crucial for the progress of DP-SGD.

## 1.1 Our contribution

In this work we present a special property of DP-SGD that is established jointly by gradient clipping and noisification. In particular, we analyze DP-SGD in a non-convex and smooth setting and identify that randomly sparsifying gradients before clipping and noisification can trigger a trade-off between internal components of the convergence bound and possibly result in a smaller upper bound. This observation suggests that in the context of DP-SGD randomly sparsified gradients could lead to faster convergence than the full gradient. This also implies that DP-SGD has inherent room for (even simply random) gradient compression. We further note that the same property is not present in other popular SGD schemes.

First, we describe the process of random sparsification (RS) and illustrate its impact on DP-SGD. RS uniformly selects a random subset $\mathcal{S}$ from the full coordinates set $\mathcal{D}$ and zeros out their gradients, we define

sparsification rate $r := |\mathcal{S}|/|\mathcal{D}|$ as the percentage of coordinates we would like to sparsify. Figure 1a, 1b schematically illustrate the vanilla DP-SGD and DP-SGD with RS using a 2D example. In this 2D example RS can either have $g_1$ or $g_2$ zeroed-out so $r$ can only be set to 0.5. In higher-dimensional cases it will be possible to set $r$ to other values within the range $(0, 1)$. Compared with vanilla DP-SGD, the drawback of RS is indeed apparent as RS removes some gradient information. However, on the other side RS has two strengths: (i) due to the isotropic Gaussian noise $\boldsymbol{\xi}_\sigma$, the amount of noise in term of Euclidean norm scales with the gradient dimension $d = |\mathcal{D}|$. With RS the effective $d$ is reduced as the sparsified coordinates do not receive any gradient information, and noisification in those dimensions is no longer necessary. Figure 1c demonstrates the empirical ratio of the noise norm with RS over without RS $\|\boldsymbol{\xi}'_\sigma\|/\|\boldsymbol{\xi}_\sigma\|$. (ii) Due to gradient clipping, for the coordinates $\mathcal{D} \setminus \mathcal{S}$ that are relaxed, i.e. corresponding gradients are not zeroed out, their gradients have a higher magnitude than without RS (see $|\bar{g}'_1| > |\bar{g}_1|$ or $|\bar{g}'_2| > |\bar{g}_2|$), implying the relaxed coordinates are better optimized, Figure 1d demonstrates an empirical ratio of the gradient norm of relaxed coordinates with RS over without RS $\|\bar{\boldsymbol{g}}'_{\mathcal{D} \setminus \mathcal{S}}\|/\|\bar{\boldsymbol{g}}_{\mathcal{D} \setminus \mathcal{S}}\|$. In Section 3, we will discuss the integrated impact of the drawback and strengthens of DP-SGD with RS for non-convex and smooth problems and as we will see RS gives rise to a trade-off between internal components of the convergence bound of DP-SGD. Another interesting observation we will present is that for other popular SGD schemes, i.e. SGD with gradient clipping (same as canceling noisification in DP-SGD), noisy SGD (same as canceling clipping in DP-SGD), and vanilla SGD, the respective convergence bounds are only enlarged by applying RS. This observation highlights the fact that the reason for RS improving DP-SGD is not trivially due to the reduced noise, and suggests that *the trade-off raised by RS in the context of DP-SGD is special, which manifests under two preconditions: gradient clipping and noisification.*

An alternative to random sparsification would be e.g. to sparsify the smallest entries of the gradient, $w_2$ in Figure 1. It is worth emphasizing that the selection of significant coordinates is also privacy related. It could be done by using a public dataset (Zhou et al., 2021; Kairouz et al., 2021) which we do not assume exists, or by providing additional privacy budget for sparse vector techniques (Dwork & Roth, 2014) or DP selection (Zhang et al., 2021), which have certain technical difficulties of defining the threshold and privacy loss for the selection, and are thus only applied when the gradient is obviously sparse, e.g. the embedding layer of a language model. Perhaps most importantly, the analysis of these methods assumes the original sparsity of the gradients (e.g. $\boldsymbol{g}$ lies in $w_1$ in Figure 1) or low-rank property (e.g. $\boldsymbol{g}$ always lies in a subspace). In contrast, our analysis, and the strengths of RS as illustrated in Figure 1, do not rely on these assumptions, which makes the contribution of this work complementary to previous works.

**Paper organization:** In Section 2, we discuss related work. In Section 3, we elaborate on the impact of RS on popular SGD schemes and finally present the special trade-off of DP-SGD induced by RS. In Section 4, we provide an efficient and lightweight RS algorithm based on our analysis. In Section 5, we discuss the additional advantages of sparsified gradients. In Section 6 we empirically verify our analysis and show the utility of our proposed algorithm. All proofs are deferred to the Appendix.

## 2 Related works

Chen et al. (2020) analyze the convergence rate of DP-SGD in a non-convex setting with characterization of the adverse effect of the gradient clipping. Many works study adaptive clipping bounds (Andrew et al., 2021; Pichapati et al., 2019). Other works study the adverse effect of noisification and prove that the performance of DP-SGD is dependent on the gradient dimension $d$ as, according to Equation (4), the amount of noise scales with $d$. Bassily et al. (2014) show that in a convex setting, DP-SGD achieves excess risk of $\tilde{O}(\sqrt{d}/n\varepsilon)$, where $n$ is the dataset size. Wang & Xu (2019) show that the empirical gradient norm decreases to $\tilde{O}(d^{1/4}/\sqrt{n\varepsilon})$ for DP-SGD with a smooth non-convex loss function.

A line of work builds on gradient space compression to improve the utility of DP-SGD. Abadi et al. (2016b) propose DP linear probing to pre-train a network on an auxiliary dataset, then transfer the feature extractor and only re-train the linear classifier on the private data. Similarly, Tramer & Boneh (2021) adopt ScatterNet (Oyallon et al., 2019) to extract handcrafted features. Both works decrease $d$ by excluding the majority of parameters during DP learning. Inspired by the empirical observation that the optimization trajectory is contained in a lower-dimensional subspace (Vogels et al., 2019; Gooneratne et al., 2020; Li et al., 2020),

several recent works (Zhou et al., 2021; Yu et al., 2021a; Kairouz et al., 2021; Yu et al., 2021b) project the gradient into a subspace which is identified by auxiliary data or released historical gradients. Zhang et al. (2021) target NLP tasks where gradients are extremely sparse, and propose a DP selection method.

In contrast to the strategy of reducing gradient space dimension, Papernot et al. (2021) propose a dedicated DP-CNN with tempered activation function which is deemed as robust to the adverse effects of gradient clipping and noisification in DP-SGD. Li et al. (2022) observe that DP-SGD works well on full fine-tuning of large language models. Furthermore, Yu et al. (2022) propose incorporating parameter-efficient fine-tuning (PEFT) techniques, such as low-rank adaptation (Hu et al., 2022) and adapter-based fine-tuning (Houlsby et al., 2019). Bu et al. (2022) propose private bias-term only fine-tuning. Both approaches improve the performance of DP fine-tuning and largely reduce the computational cost with large models, as well as the communication overhead in distributed settings. However, PEFT methods require a pre-trained model as a backbone and cannot be applied in scenarios where the network must be trained from scratch.

The studies of Damaskinos et al. (2021) and Mangold et al. (2022) on DP coordinate descent compute the gradient of a random coordinate in backpropagation and in this aspect similar to RS. However, there is notable difference between these existing works and our work. Damaskinos et al. (2021) study the generalized linear model which includes a convex problem and investigate the dual formulation. Mangold et al. (2022) also analyze convex optimization problem and rely on precise coordinate-wise regularity measures of the objective function such that each coordinate can be noisified accordingly. Both works derive a modified update step (beside sparsification). In this work, we study DP-SGD with RS in a non-convex setting. We do not assume detailed characterization of the optimization landscape, e.g. coordinate-wise smoothness. And RS is applied directly on the gradient computed with the loss function of the network's prediction. Our analysis is thereby more general.

## 3 Applying random sparsification to gradient descent methods

**Notation & Terminology**   Let $\mathcal{L}$ be the objective function $\mathcal{L}(w) := \mathbb{E}_{x \in \mathcal{X}}[\ell(w; x)]$ which is $G$-Lipschitz smooth, and $x$ denotes a training example sampled from the dataset $\mathcal{X}$. We define $g_{t,i} := \nabla \ell(w_t; x_i)$ the gradient at step $t$ with example $x_i$ and define the true gradient $\nabla w_t := \mathbb{E}_{x \in \mathcal{X}}[g_{t,i}]$. We assume the gradient deviation $g_{t,i} - \nabla w_t$ is sampled from a zero-mean random variable $\xi_t$. The averaged gradient of $B$ samples at the step $t$ is denoted as $g_t := \frac{1}{B} \sum_i g_{t,i}$, and the averaged clipped gradient as $\bar{g}_t := \frac{1}{B} \sum_i g_{t,i} \cdot \min(1, C/\|g_{t,i}\|)$. To conduct RS, we use a random mask $m \in \{0, 1\}^d$ and uniformly draw $rd$ indices and set these positions in the mask to 0 while others to 1 so that $\|m\|_1 = (1 - r)d$, $r$ is the predefined sparsification rate. The average of sparsified and clipped gradients can thus be expressed as $\hat{g}_t := \frac{1}{B} \sum_i g'_{t,i} \cdot \min(1, C/\|g'_{t,i}\|)$, where we define $g'_{t,i} := m \odot g_{t.i}$, the Hadamard product is denoted using $\odot$. For noisification, isotropic Gaussian noise is denoted as $\xi_\sigma$, when RS is applied we do $m \odot \xi_\sigma$. As this work involves multiple ways of processing gradients, we summarize their notation in Table 1 for convenience. We do not bold vectors or matrices in the analysis since it is clear from the context. As RS modifies DP-SGD, we clarify that RS does not breach privacy with the following theorem:

**Theorem 3.1.** *For a Gaussian mechanism: $\mathcal{M}(x) := \sum_{x \in X} g(x) \cdot \min(1, C/\|g(x)\|) + \mathcal{N}(0, C^2 \sigma^2 \mathbf{I}_d)$, which satisfies $(\varepsilon, \delta)$-DP, after applying RS with a mask $m \in \{0, 1\}^d$, the modified Gaussian mechanism $\mathcal{M}'(x) := \sum_{x \in X} m \odot g(x) \cdot \min(1, C/\|m \odot g(x)\|) + m \odot \mathcal{N}(0, C^2 \sigma^2 \mathbf{I}_d)$ also satisfies $(\varepsilon, \delta)$-DP.*

*Proof.* Note that $\mathcal{M}'$ is equivalent to $\mathcal{M}''(x) := \sum_{x \in X} g_{\mathcal{D} \setminus \mathcal{S}}(x) \cdot \min(1, C/\|g_{\mathcal{D} \setminus \mathcal{S}}(x)\|) + \mathcal{N}(0, C^2 \sigma^2 \mathbf{I}_{|\mathcal{D} \setminus \mathcal{S}|})$ in terms of privacy, then it is easy to observe that $\mathcal{M}''(x)$ and $\mathcal{M}(x)$ provide the same level of DP. It should be noted that the scenario of $m = \mathbf{0}$ is not considered in our analysis as it does not align with the objectives of optimization. $\square$

### 3.1 Applying random sparsification to vanilla SGD and noisy SGD

We first note that for vanilla SGD (Bottou et al., 2018), applying RS is the same as randomly dropping gradient information, thus RS cannot provide any benefits. Next, we consider the impact of RS under noisification and gradient clipping separately. If we cancel the gradient clipping in DP-SGD, the resulting optimization

| Gradient | |
|---|---|
| $g_{t,i}$ | Gradient at step $t$ with example $x_i$, i.e. $\nabla \ell(w_t, x_i)$. |
| $\nabla w$ | True gradient at step $t$, i.e. $\mathbb{E}_{x \in \mathcal{X}}[g_{t,i}]$ |
| $g_t$ | Mini-batch gradient at step $t$, i.e. $\frac{1}{B} \sum_i g_{t,i}$ |
| $\bar{g}_t$ | Clipped mini-batch gradient at step $t$, i.e. $\frac{1}{B} \sum_i g_{t,i} \cdot \min(1, C/\|g_{t,i}\|)$ |
| $g'_{t,i}$ | Sparsified gradient due to random mask $m$, i.e. $m \odot g_{t,i}$. |
| $\nabla w'$ | Sparsified true gradient due to random mask $m$, i.e. $m \odot \mathbb{E}_{x \in \mathcal{X}}[g_{t,i}]$ |
| $\hat{g}_t$ | Sparsified and clipped mini-batch gradient at step $t$, i.e. $\frac{1}{B} \sum_i g'_{t,i} \cdot \min(1, C/\|g'_{t,i}\|)$. |
| Gradient deivation | |
| $\xi_t$ | Assumed random variable generating gradient deviations, i.e. $g_{t,i} - \nabla w_t$. |
| $\xi'_t$ | Assumed random variable generating sparsified gradient deviations, i.e. $g'_{t,i} - \nabla w'_t$. |
| $p_t$ | True probability distribution of $\xi_t$. |
| $\tilde{p}_t$ | A symmetric proxy of $p_t$ which approximates $p_t$ while satisfies $\tilde{p}_t(\xi_t) = \tilde{p}_t(-\xi_t)$. |
| $p'_t$ | True probability distribution of $\xi'_t$. |
| $\tilde{p}'_t$ | A symmetric proxy of $p'_t$ which is projected from $p_t$ by random mask $m$. |

Table 1: Notations for gradients.

method is noisy SGD which has been used in Bayesian learning, e.g. stochastic gradient Langevin dynamics (Welling & Teh, 2011). When RS is applied (pseudo code is given in Appendix E), the amount of noise will be reduced as illustrated in Figure 1, however such a benefit seems insufficient to compensate the loss of gradient information, which we show in Theorem 3.2.

**Theorem 3.2.** *Consider noisy SGD with RS on a $G$-smooth function $\mathcal{L}$ with isotropic Gaussian noise $\xi_\sigma \sim \mathcal{N}(0, \sigma^2 \boldsymbol{I}_d)$, learning rate $\gamma = \frac{1}{G\sqrt{T}}$, batch size $B$ and sparsification rate $r$, assume an upper bound of individual gradient deviation $\|\xi_t\|^2 \leq \sigma_g^2$, we have:*

$$\frac{1}{T} \sum_{t=1}^{T} \|\nabla w_t\|^2 \leq \frac{2G}{(1-r)\sqrt{T}} \Delta_{\mathcal{L}} + \frac{1}{\sqrt{T}} \left( \frac{\sigma_g^2}{B} + \frac{d\sigma^2}{B^2} \right), \tag{5}$$

where we define $\Delta_{\mathcal{L}} := \mathbb{E}[\mathcal{L}(w_1)] - \min_w \mathcal{L}(w)$. We note that the noise term $d\sigma^2/B^2$ on the r.h.s. was reduced by a factor of $1 - r$ due to the RS, but this factor is canceled out by the same factor from the l.h.s. (see Appendix A). Additionally, Equation (5) can also be used to describe the case of vanilla SGD with RS by removing the last term $d\sigma^2/B^2$ at the r.h.s. Overall, we have the following observation:

---

**Remark 1.** Theorem 3.2 suggests that in applying RS to noisy SGD or vanilla SGD, the network may converge more slowly, as $\|\nabla w_t\|^2$ has a larger upper bound for $0 < r < 1$.

---

## 3.2 Applying random sparsification to SGD with gradient clipping

If we cancel the noisification in DP-SGD, then the optimization method becomes SGD with gradient clipping, which has been widely adopted for large network training to prevent exploding gradients or stabilize optimization (Zhang et al., 2020; Pascanu et al., 2013). If RS is applied (pseudo code is given in Appendix E), relaxed coordinators can be better optimized as illustrated in Figure 1, however such benefit also seems insufficient to compensate the loss of gradient information.

First consider the convergence of gradient clipping without sparsification.

**Lemma 3.1.** *Consider SGD on a $G$-smooth function $\mathcal{L}$ with gradient clipping of bound $C$, learning rate $\gamma$, we have:*

$$\mathbb{E}[\langle \nabla w_t, \bar{g}_t \rangle] \leq \frac{1}{\gamma} \mathbb{E}[\mathcal{L}_t - \mathcal{L}_{t+1}] + \gamma \frac{GC^2}{2}. \tag{6}$$

As the distribution of the gradient is unknown, the expectation involving gradient clipping, i.e. $\mathbb{E}[\langle \nabla w_t, \bar{g}_t \rangle]$, impedes the estimation of the convergence rate. Let $p_t$ be the true distribution of the gradient deviation $\xi_t$.

Chen et al. (2020) firstly observe that $p_t$ appears to be symmetric and use this property to relate $\mathbb{E}[\langle \nabla w_t, \bar{g}_t \rangle]$ to $\|\nabla w_t\|$:

**Lemma 3.2** (Chen et al. (2020)). *Assume $p_t(\xi_t) = p_t(-\xi_t)$, $\forall \xi_t \in \mathbb{R}^d$, gradient clipping with bound $C$ has the following property:*

$$\mathbb{E}[\langle \nabla w_t, \bar{g}_t \rangle] \geq P_{\xi_t \sim p_t}(\|\xi_t\| < 3C/4)h(\nabla w)\|\nabla w_t\|, \tag{7}$$

where $h(\nabla w) := \min(\|\nabla w_t\|, C/4)$.

We see the r.h.s. of Equation (7) is proportional to $\|\nabla w\|$ or $\|\nabla w\|^2$ as long as $P_{\xi_t \sim p_t}(\|\xi_t\| < 3C/4)$ is not close to zero. Combining Lemma 3.1 with Lemma 3.2, it is possible to form an upper bound for finding the critical point. However, the real gradient distribution cannot be exactly symmetric. Instead of using $p_t$ in Equation (7), we can choose a proxy $\tilde{p}$ which is symmetric and use an error term $b_t$ to represent the difference as $b_t := \mathbb{E}_{\xi_t \sim p_t}[\langle \nabla w, \bar{g}_t \rangle] - \mathbb{E}_{\xi_t \sim \tilde{p}_t}[\langle \nabla w, \bar{g}_t \rangle]$.

**Theorem 3.3** (Chen et al. (2020)). *Consider SGD on a $G$-smooth function $\mathcal{L}$ with gradient clipping bound $C$ and learning rate $\gamma$. Choose a symmetric distribution $\tilde{p}(\cdot)$ satisfying $\tilde{p}_t(\xi_t) = \tilde{p}_t(-\xi_t)$, $\forall \xi_t \in \mathbb{R}^d$, and consider $T$ iterations:*

$$\frac{1}{T}\sum_{t=1}^{T} P_{\xi_t \sim \tilde{p}_t}(\|\xi_t\| < 3C/4)h(\nabla w_t)\|\nabla w_t\| \leq \frac{\Delta_{\mathcal{L}}}{\gamma T} + \gamma\frac{GC^2}{2} - \frac{1}{T}\sum_{t=1}^{T} b_t. \tag{8}$$

Chen et al. (2020) argues that there exists $\tilde{p}_t$ which is a close approximation of $p_t$ such that $b_t$ is small while $P_{\xi_t \sim \tilde{p}_t}(\|\xi_t\| < 3C/4)$ is bounded away from zero, as in practice $p_t$ tends to be approximately symmetric. They also empirically demonstrate that $p_t$ becomes approximately symmetric during training. So Theorem 3.3 indicates the convergence rate of SGD with gradient clipping. To make sure this property of $p_t$ is generally valid in our case, we have also verified that $p_t$ becomes approximately symmetric in our experimental environments in Section 6 (see Appendix F).

Now consider the convergence rate after applying RS, note that the sparsification happens before clipping, so the conclusion cannot be straightforwardly derived from Theorem 3.3. Since $m$, $\xi_t$ are independent:

$$\mathbb{E}[\langle \nabla w_t, \hat{g}_t \rangle] = \mathbb{E}_m\left[\mathbb{E}_{\xi'_t \sim \tilde{p}'_t}[\langle \nabla w'_t, \hat{g}_t \rangle]\right] + \mathbb{E}_m[b'_t], \tag{9}$$

where we define $\nabla w'_t := m \odot \nabla w_t$, $\xi'_t := m \odot \xi_t$ with corresponding true distribution $p'_t$ and proxy $\tilde{p}'_t$ which are projected from $p_t$ and $\tilde{p}_t$, $b'_t := \mathbb{E}_{\xi'_t \sim p'_t}[\langle \nabla w', \hat{g}_t \rangle] - \mathbb{E}_{\xi'_t \sim \tilde{p}'_t}[\langle \nabla w', \hat{g}_t \rangle]$. Since the projection of a symmetric distribution to a subspace is symmetric, we have $\tilde{p}'_t(\xi'_t) = \tilde{p}'_t(-\xi'_t)$, so following Lemma 3.2, we have for the first term on the r.h.s. of Equation (9):

$$\mathbb{E}_m\left[\mathbb{E}_{\xi'_t \sim \tilde{p}'_t}[\langle \nabla w'_t, \hat{g}_t \rangle]\right] \geq \mathbb{E}_m[P_{\xi'_t \sim \tilde{p}'_t}(\|\xi'_t\| < 3C/4)h(\nabla w'_t)\|\nabla w'_t\|]. \tag{10}$$

Then to solve the expectation over the random mask $m$ in the Equation (10), we provide Lemma 3.3:

**Lemma 3.3.** *Apply RS with sparsification rate $r$, choose $\tilde{p}_t$ such that $P_{\xi_t \sim \tilde{p}_t}(\|\xi_t\| < 3C/4) \geq \sqrt{1-r}$, then $\exists \kappa_t \in (1-r, 1)$ such that:*

$$\mathbb{E}_m[P_{\xi'_t \sim \tilde{p}'_t}(\|\xi'_t\| < 3C/4)h(\nabla w'_t)\|\nabla w'_t\|] = \kappa_t P_{\xi_t \sim \tilde{p}_t}(\|\xi_t\| < 3C/4)h(\nabla w_t)\|\nabla w_t\|, \tag{11}$$

$\kappa_t$ takes the value 1 when $r = 0$. Based on Lemma 3.3, we provide Theorem 3.4 to characterize the convergence of SGD with gradient clipping and RS.

**Theorem 3.4.** *Consider SGD on a $G$-smooth function $\mathcal{L}$ with gradient clipping of bound $C$, learning rate $\gamma$, and apply RS with sparsification rate $r$. Choose a symmetric distribution $\tilde{p}(\cdot)$ satisfying $\tilde{p}_t(\xi_t) = \tilde{p}_t(-\xi_t)$, $\forall \xi_t \in \mathbb{R}^d$, while $P_{\xi_t \sim \tilde{p}_t}(\|\xi_t\| < 3C/4) \geq \sqrt{1-r}$, and consider $T$ iterations, then $\exists \kappa \in (1-r, 1)$ s.t.:*

$$\frac{1}{T}\sum_{t=1}^{T} P_{\xi_t \sim \tilde{p}_t}(\|\xi_t\| < 3C/4)h(\nabla w_t)\|\nabla w_t\| \leq \frac{1}{\kappa}\left(\frac{\Delta_{\mathcal{L}}}{\gamma T} + \gamma\frac{GC^2}{2} - \frac{1}{T}\sum_{t=1}^{T} \mathbb{E}_m[b'_t]\right), \tag{12}$$

**Remark 2.** Consider that $b_t$, $b'_t$ tend to be small and negligible, compared with Theorem 3.3, Theorem 3.4 suggests that RS could impede the convergence of SGD with gradient clipping as $1/\kappa > 1$ for $r > 0$.

### 3.3 Applying random sparsification to DP-SGD

Although there is seemingly no advantage of using RS for noisy SGD or SGD with gradient clipping, we show that in case of DP-SGD, i.e. when noisification and gradient clipping are both presented, RS can induce a trade-off between internal components of the convergence bound and possibly achieve a smaller upper bound.

**Lemma 3.4.** *Consider DP-SGD on a G-smooth function $\mathcal{L}$ with clipping bound $C$, isotropic Gaussian noise $\xi_\sigma \sim \mathcal{N}(0, \sigma^2 C^2 \boldsymbol{I}_d)$, learning rate $\gamma$, batch size $B$, sparsification rate $r$, we have:*

$$\mathbb{E}[\langle \nabla w_t, \hat{g}_t \rangle] \leq \frac{1}{\gamma} \mathbb{E}[\mathcal{L}_t - \mathcal{L}_{t+1}] + \gamma \frac{GC^2}{2} + (1-r)\gamma\Delta_\sigma, \tag{13}$$

where we have defined $\Delta_\sigma := \frac{C^2 \sigma^2 dG}{2B^2}$. Now we provide Theorem 3.5 to characterize the convergence rate of DP-SGD with RS.

**Theorem 3.5.** *Consider DP-SGD on a G-smooth function $\mathcal{L}$ with clipping bound $C$, isotropic Gaussian noise $\xi_\sigma \sim \mathcal{N}(0, \sigma^2 C^2 \boldsymbol{I}_d)$, learning rate $\gamma$, batch size $B$, and apply RS with sparsification rate $r$. Choose a symmetric distribution $\tilde{p}(\cdot)$ satisfying $\tilde{p}_t(\xi_t) = \tilde{p}_t(-\xi_t), \forall \xi_t \in \mathbb{R}^d$, while $P_{\xi_t \sim \tilde{p}_t}(\|\xi_t\| < 3C/4) \geq \sqrt{1-r}$, and consider $T$ iterations, then $\exists \kappa \in (1-r, 1)$, such that:*

$$\frac{1}{T}\sum_{t=1}^{T} P_{\xi_t \sim \tilde{p}_t}(\|\xi_t\| < 3C/4)h(\nabla w_t)\|\nabla w_t\| \leq \frac{1}{\kappa}\left(\frac{\Delta_\mathcal{L}}{\gamma T} + \gamma\frac{GC^2}{2} - \frac{1}{T}\sum_{t=1}^{T}\mathbb{E}_m[b'_t]\right) + \frac{1-r}{\kappa}\gamma\Delta_\sigma. \tag{14}$$

It is worth noting that in Theorem 3.5 on the r.h.s. the noise term $\Delta_\sigma$ has a factor $1 - r$ which is induced by sparsification, but unlike under noisy SGD (see Theorem 3.2), this factor is not fully canceled out as $\kappa > 1 - r$. Also note that with no sparsification ($r = 0$), we have $\kappa = 1$, Theorem 3.5 has the following form:

$$\frac{1}{T}\sum_{t=1}^{T} P_{\xi_t \sim \tilde{p}_t}(\|\xi_t\| < 3C/4)h(\nabla w_t)\|\nabla w_t\| \leq \frac{\Delta_\mathcal{L}}{\gamma T} + \gamma\left(\frac{GC^2}{2} + \Delta_\sigma\right) - \frac{1}{T}\sum_{t=1}^{T} b_t, \tag{15}$$

which recovers the convergence bound of vanilla DP-SGD (Chen et al., 2020), implying that our convergence bound over RS is as tight as before. Comparing Equation (15) with Theorem 3.5, we observe:

> **Remark 3.** RS introduces two factors on the r.h.s.: $1/\kappa, (1-r)/\kappa$. The noise term $\Delta_\sigma$ is reduced by $(1-r)/\kappa$ while the remaining terms are enlarged by by $1/\kappa$: a trade-off is established. The respective magnitude of $\Delta_\sigma$ and other terms determines whether it is a gain or loss to apply RS. It is worthwhile to note that $\Delta_\sigma$ scales with $C^2\sigma^2$ and the gradient dimension $d$, and is inversely proportional to $B^2$.

We also present the upper bound using privacy budget variables $(\varepsilon, \delta)$ in Appendix C with Corollary C.1.1.

## 4 Implementation of random sparsification for DP-SGD

Theorem 3.5 reveals that RS induces a trade-off between the internal components of the convergence bound and has a chance to accelerate the convergence. However, the best sparsification rate is infeasible to be computed *a priori* and possibly varying during optimization. In this section, we present a practically efficient and lightweight RS approach. *The additional cost of running RS is negligible.*

Algorithm 1 outlines our approach. RS is also compatible with SGD with gradient momentum and Adam (Kingma & Ba, 2014). In case of non-zero momentum, coordinates that have been selected can still be updated as long as their velocity has not decayed to zero. To make the algorithm efficient in practice, we adopt gradual cooling (Line 3) and per-epoch randomization (Line 5), which we discuss in the sequel.

### 4.1 Gradual cooling

In practice we find that if we initiate the training with a constant large sparsification rate $r$, the network converges slowly and performs poorly when the privacy budget has been fully consumed. According to Theorem 3.5 and Remark 3, we see sparsification is beneficial once $\Delta_\sigma$ is significant compared with the other

---

**Algorithm 1** DP-SGD with Random Sparsification

---

**Input:** Initial parameters $w_0$; Epochs $E$; Sparsification rate: $r^*$; Clipping bound: $C$; Noise multiplier $\sigma$; Momentum: $\mu$; Learning rate $\gamma$.

1: **for** $e = 0$ to $E - 1$ **do**
2:     ▷ *Gradual cooling* ◁
3:     $r(e) = r^* \cdot \frac{e}{E-1}$;
4:     ▷ *Generate a random mask every epoch* ◁
5:     $m \in \{0,1\}^d$, s.t. $\|m\|_1 = d \cdot (1 - r(e))$;
6:     **for** $t = 0$ to $T - 1$ **do**
7:        ▷ *For each $x_i$ in the Poisson-sampled batch $\mathcal{B}$* ◁
8:        $g_{t,i} = \nabla \ell(w_t, x_i)$;
9:        ▷ *Sparsify gradient* ◁
10:        $g'_{t,i} = m \odot g_{t,i}$;
11:        ▷ *Clip each individual gradient* ◁
12:        $\hat{g}_t = \frac{1}{|\mathcal{B}|} \sum_{i \in \mathcal{B}} g'_{t,i} \cdot \min(1, C/\|g'_{t,i}\|)$;
13:        ▷ *Add sparsified noise* ◁
14:        $\tilde{g}_t = \hat{g}_t + m \odot \mathcal{N}(0, \frac{C^2 \sigma^2}{|\mathcal{B}|^2} \boldsymbol{I}_d)$;
15:        ▷ *Update parameters* ◁
16:        $v_{t+1} = \mu \cdot v_t + \tilde{g}_t, w_{t+1} = w_t - \gamma v_{t+1}$;

---

terms in the convergence bound. At early training stages the network converges fast: $\mathbb{E}[\mathcal{L}_t - \mathcal{L}_{t+1}]$ is large, while during training the optimization reaches a plateau: $\mathbb{E}[\mathcal{L}_t - \mathcal{L}_{t+1}]$ decays. In contrast, $\Delta_\sigma$ is always fixed and therefore its relative significance grows progressively. To fit this dynamics, we use a simple gradual cooling strategy which linearly ramps up the sparsification rate from 0 to $r^*$ during training, i.e. $r = r^* \cdot \frac{e}{E-1}$. We provide a more detailed discussion in Appendix G.

### 4.2 Per-epoch randomization

RS alternates the optimization direction. As SGD takes several steps to reach the local minimum, frequently alternating optimization direction might be harmful, which we have also observed in the experiments. Considering that tuning over the number of iterations for refreshing the random mask can be expensive, in this work we set the refresh to be per-epoch. Another strength of per-epoch randomization is that for one epoch there are certainly $(1-r)d$ coordinates updated, which is favorable in distributed learning as communication overhead is a key issue, and data are not transmitted every iteration. For per-iteration randomization, the cumulative number of updated parameters depends on the sparsification rate $r$ and iterations between two communication rounds, resulting in higher communication overheads than per-epoch randomization.

## 5 Advantages of sparsified gradients

Sparse representation of the gradient generally offers additional advantages. Our analysis in Section 3 demonstrates that DP-SGD provides an inherent potential for trivial random gradient compression. In the sequel, we discuss the benefits of sparsified gradients in terms of private machine learning.

**Reduced communication overhead of DP federated learning**   A line of work studies how to incorporate differential privacy in federated learning, which is a distributed learning scheme with the interest of protecting the privacy of participants (McMahan et al., 2017; Yang et al., 2019; Shokri & Shmatikov, 2015; Liu et al., 2020). While a major issue in federated learning is the communication bottleneck, sparse representation produced by RS can be transferred in form of non-zero values and indices. The cost of indexing is logarithmic in the number of parameters. Considering that $log_2 10^9 < 32$ and for per-epoch randomization multiple rounds of communication could share the same indices, the cost of indexing is negligible, and communication overhead by RS is reduced to $\tilde{O}(1 - r)$.

**Strengthened privacy against gradient reconstruction attacks** A gradient reconstruction attack is one of the most hazardous privacy attacks, which assumes that the adversary has access to victim's gradient and intends to recover the victim's training data through gradient matching (Zhu et al., 2019; Geiping et al., 2020; Zhu & Blaschko, 2021; Yin et al., 2021; Zhu et al., 2023). This is a common scenario in distributed learning schemes, as participants need to share their local update when computing the global model. Zhu & Blaschko (2021) provide a closed-form solution of a gradient reconstruction attack: for a certain layer one can form a constraint matrix $K$ over the input $x$ given $g$, so that $Kx = g$, where $x, g$ are flattened vectors (Zhu & Blaschko, 2021, Equation (15)). When $K$ is overdetermined, after receiving the gradient $g$, the input $x$ can be reconstructed through least squares: $x = (K^\top K)^{-1} K^\top g$. To prevent such an attack, DP-SGD can be introduced so that $g$ will be perturbed by Gaussian noise $\xi_\sigma$, which leads to a squared reconstruction error:

$$\|x - \hat{x}\|^2 = \|(K^\top K)^{-1} K^\top \xi_\sigma\|^2. \tag{16}$$

If we add more noise, the expected squared error $\mathbb{E}[\|x - \hat{x}\|^2]$ will be increased, but at the cost that the network will lose performance. However, under a certain noise level, applying RS can further amplify the reconstruction error (although the noise will also be sparsified).

To demonstrate this, we consider a single layer network with a one digit input $x \in \mathbb{R}$, a constraint vector $k \in \mathbb{R}^d$ and corresponding gradient vector $g \in \mathbb{R}^d$:

**Theorem 5.1.** *Conduct an attack using Equation* (16) *on the target data* $x$, *consider that the gradient* $g$ *is perturbed by isotropic Gaussian noise* $\xi_\sigma \sim \mathcal{N}(0, \sigma^2 \boldsymbol{I}_d)$ *and further sparsified with sparsification rate* $r \in (0, 1)$. *The expected squared error of reconstruction is:*

$$\mathbb{E}[\|x - \hat{x}\|^2] \geq \frac{\sigma^2}{(1 - r)\|k\|^2}. \tag{17}$$

We see the expected squared error has a lower bound which increases monotonically with the sparsification rate $r$. We also verify this phenomenon under a more general attacking scenario in Section 6.3.

# 6 Experiments

**Setup:** Our code is implemented in PyTorch (Paszke et al., 2019b). To compute the gradients of an individual example in a mini-batch we use the BackPACK package (Dangel et al., 2020). Cumulative privacy loss has been tracked with the Opacus package, which adopts Rényi differential privacy (Mironov, 2017; Balle et al., 2020). A Poisson-sampled batch of data at each iteration implies privacy amplification (Balle et al., 2018; Wang et al., 2019; Mironov et al., 2019). In the implementation of many previous works (Papernot et al., 2021; Tramer & Boneh, 2021; Yu et al., 2021a), sampling is conducted by randomly shuffling and partitioning the dataset into batches of fixed size, we follow this convention. We focus on DP image classification and run all experiments on a cluster within the same container environment 5 times using the same group of 5 random seeds. Our code is available at `https://github.com/JunyiZhu-AI/RandomSparsification`.

## 6.1 Evidence of the trade-off and its uniqueness for DP-SGD

Theorem 3.5 implies that when the noise term $\Delta_\sigma$ is dominant in the convergence bound, the trade-off induced by RS can achieve a smaller upper bound on the convergence rate. In the context of DP training, the number of iterations is limited, so faster convergence means better accuracy. Since $\Delta_\sigma \propto \sigma^2 C^2$, we conduct experiments on various combinations of $\sigma$ and $C$ to verify our analysis. When conducting RS, we always adopt gradual cooling as discussed in Section 4.1 to reach the final sparsification rate. Based on Figure 2, except the last Figure 2i, where due to excessive noise the network does not converge after a few training epochs, we see: (i) from left to right, as $\sigma$ goes up, more RS is preferred; (ii) from top to bottom, as $C$ increases, more can be gained from RS; (iii) the trade-off has the same tendency for the same value of $\sigma^2 C^2$. These observations verify our analysis. We note that as $C^2\sigma^2$ increases to $2^2$, i.e. Figure 2f and 2h, RS can improve the performance of DP-SGD by more than 4%. Additionally, when $C^2\sigma^2 = 1$, i.e. Figure 2c, 2e, 2g, The performance is maintained under RS (or slightly increased). It should be noted that

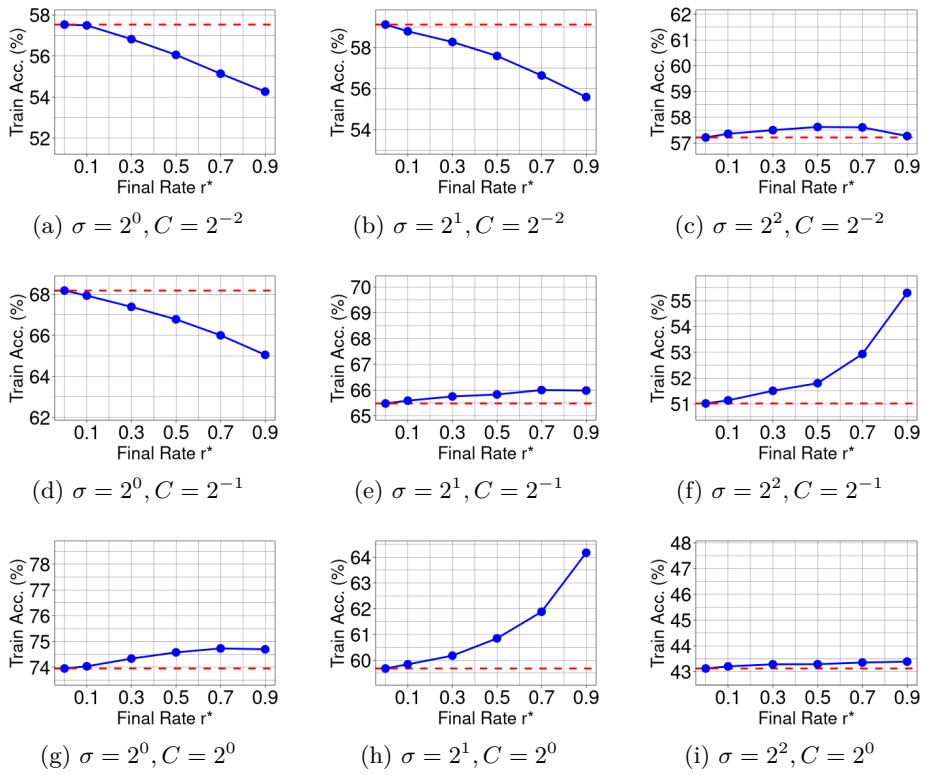

Figure 2: Train accuracy vs. final sparsification rate $r^*$ over various combinations of noise multiplier $\sigma$ and clipping bound $C$, red dashed line marks the performance of no sparsification. We set batch size to 1000 and train for 100 epochs, the network is DP-CNN. The result shows that the trade-off favors RS as $\Delta_\sigma \propto \sigma^2 C^2$ increasing, except the last Figure 2i, where due to excessive noise the network degrades after a few epochs.

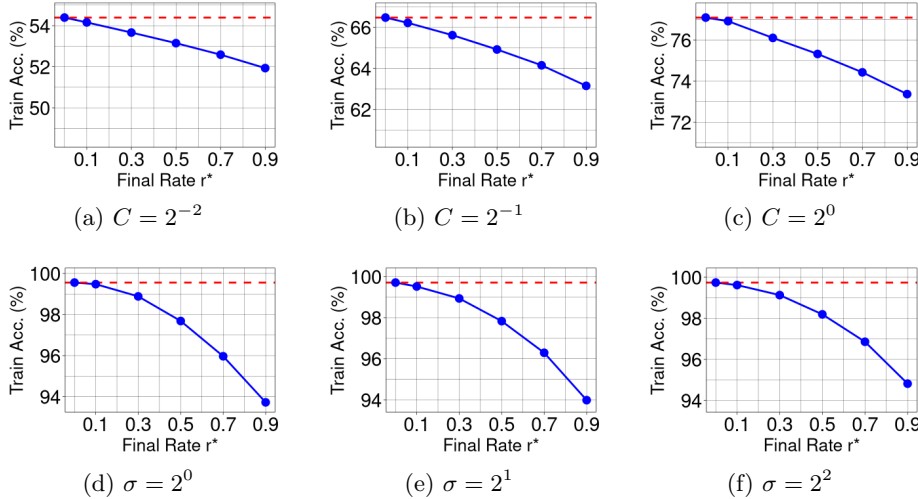

Figure 3: Train accuracy vs. final sparsification rate $r^*$ over SGD with gradient clipping or noisy SGD (see Algorithms 2 and 3 for pseudo code), other settings are the same as Figure 2. The red dashed line marks the performance of no sparsification. Without noisification or gradient clipping RS only leads to utility drop, which matches our theoretical analysis and indicates that the trade-off is a unique property of DP-SGD.

| Dataset | Approaches | $\varepsilon$ | $d$ | Baseline | RS (ours) | Difference |
|---------|-----------|-----|-----|----------|-----------|------------|
| CIFAR10 | H-CNN | 3.0 | 187K | $69.7 \pm 0.19$ | $\mathbf{70.0 \pm 0.11}$ | +0.3 |
|         |       | 1.0 |      | $62.4 \pm 0.13$ | $\mathbf{63.2 \pm 0.10}$ | +0.8 |
|         | DP-CNN | 3.0 | 550K | $62.8 \pm 0.10$ | $\mathbf{64.3 \pm 0.17}$ | +1.5 |
|         |       | 1.0 |      | $52.5 \pm 0.25$ | $\mathbf{55.1 \pm 0.11}$ | +2.6 |
| SVHN    | H-CNN | 3.0 | 187K | $85.9 \pm 0.06$ | $\mathbf{86.8 \pm 0.13}$ | +0.9 |
|         |       | 1.0 |      | $80.7 \pm 0.15$ | $\mathbf{82.2 \pm 0.19}$ | +1.5 |
|         | DP-CNN | 3.0 | 550K | $83.4 \pm 0.11$ | $\mathbf{84.5 \pm 0.14}$ | +1.1 |
|         |       | 1.0 |      | $76.0 \pm 0.05$ | $\mathbf{79.1 \pm 0.13}$ | +3.1 |
| FMNIST  | H-CNN/S | 3.0 | 33K | $88.9 \pm 0.09$ | $\mathbf{89.2 \pm 0.07}$ | +0.3 |
|         |       | 1.0 |      | $85.8 \pm 0.17$ | $\mathbf{87.0 \pm 0.08}$ | +1.2 |
|         | DP-CNN/S | 3.0 | 26K | $86.6 \pm 0.09$ | $\mathbf{87.4 \pm 0.10}$ | +0.8 |
|         |       | 1.0 |      | $83.2 \pm 0.10$ | $\mathbf{84.5 \pm 0.16}$ | +1.3 |

Table 2: Test accuracy ($\% \pm$ SEM) before and after adopting random sparsification. The difference of mean accuracy is presented in the last column.

such a flat trend is also desirable as sparse gradients produced by RS are beneficial for private learning as discussed in Section 5.

Theorems 3.2 and 3.4 suggest that the trade-off of DP-SGD is jointly established by noisification and clipping, as with either absent, RS just reduces performance. To verify this, we conduct experiments on two ablation variants of DP-SGD: noisy SGD and SGD with gradient clipping (pseudo code is given in Appendix E). Figure 3 demonstrates that in the absence of either noisification or gradient clipping, RS always makes the convergence slower, which indicates that the trade-off is a unique property of DP-SGD.

## 6.2 Improving performance of DP networks

In practice, the clipping bound $C$ and noise multiplier $\sigma$ are tuned as hyperparameters for the networks to achieve the best performance under a certain privacy budget. To investigate whether RS has a chance to achieve better performance in practical settings, we conduct experiments on the following baselines of DP image classification: (i) DP-CNN (Papernot et al., 2021): a network for training from scratch; (ii) H-CNN (Tramer & Boneh, 2021): a network for training solely on a private dataset, which uses handcrafted features. Both are representative SOTA frameworks in high privacy regimes (De et al., 2022). We adopt the best hyperparameters provided in previous works, then we do grid search for baselines and RS over the clipping bound $C \in \{0.1, 0.5, 1\}$ where $C = 0.1$ is given in previous works, and epochs $E \in \{E^*, 1.2 \cdot E^*, 1.5 \cdot E^*\}$ ($\sigma$ is adapted accordingly), where $E^*$ is the best given in previous works for different frameworks. When conducting RS we use gradual cooling and search for the final sparsification rate $r^* \in \{0.5, 0.7, 0.9\}$. We also find an interesting scaling rule to efficiently find the optimal hyperparameters for DP frameworks which is given in Appendix H. As a result, some baseline performances we give are higher than reported in previous works. As in the previous section, we observe that RS is beneficial when $\Delta_\sigma$ is dominant, thus we conduct experiments at high privacy regimes, i.e. $\varepsilon \in \{1, 3\}$.

Table 2 shows that RS improves the performance of baselines and the gain is significant for smaller $\varepsilon$, where more noise is added. We further note that DP-CNN generally gains more from RS, as it has comparatively more parameters and $\Delta_\sigma$ scales with the gradient dimension $d$.

## 6.3 Advantages of sparse gradients

Beyond the improvement in performance, sparse gradients also offer additional advantages in the area of private machine learning as discussed in Section 5. To investigate the sparsity we can achieve, we directly plugged RS into various frameworks, including: DP-TL (Tramer & Boneh, 2021): a transfer learning framework, and GEP (Yu et al., 2021a): a projected DP-SGD framework, as well as DP-CNN and H-CNN, using the hyperparameters of no sparsification given in previous works (so the same iterations). Figure 4 shows

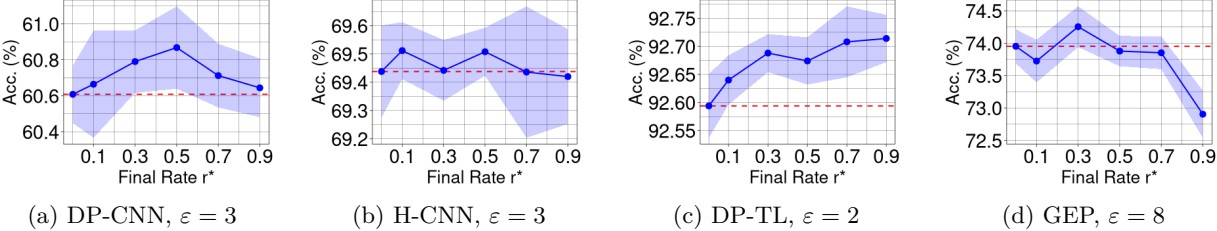

(a) DP-CNN, $\varepsilon = 3$      (b) H-CNN, $\varepsilon = 3$      (c) DP-TL, $\varepsilon = 2$      (d) GEP, $\varepsilon = 8$

Figure 4: Test accuracy ($\% \pm$ SEM) vs. final sparsification rate $r^*$ when random sparsification is directly plugged into different frameworks using the hyperparameters of no sparsification given in previous works.

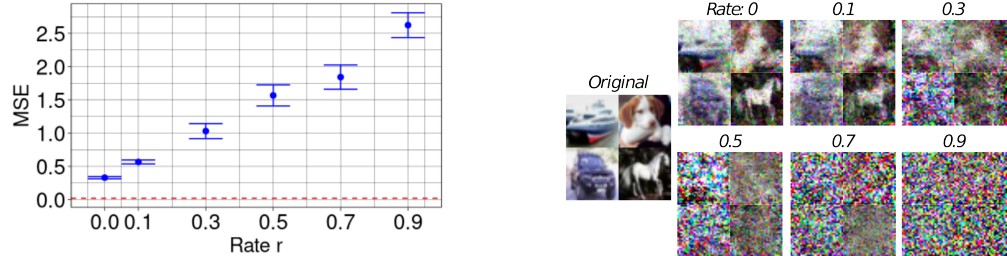

Figure 5: MSE of reconstruction using IG (Geiping et al., 2020) vs. sparsification rate $r$. In the non-private learning scheme, the MSE is $0.02 \pm 0.02$ represented by the red dashed line. The victim's network is ResNet34 and added noise is sampled from $\mathcal{N}(0, 10^{-3})$. The results are computed on 200 images from CIFAR10, four example images are: ship, dog, car, horse.

that we can achieve up to 0.7 or 0.9 final sparsification rate, i.e. 0.35 or 0.45 average sparsity under gradual cooling, while maintaining the same performance.

In Section 5 we have theoretically proven that with sparse representation, the communication overhead of DP federated learning improves. We also provide Theorem 5.1 to show that the sparse gradients produced by RS strengthen the privacy against gradient reconstruction attacks through an example of reconstruction using the linear relation between data and gradient. To verify this advantage under a general setting, we conduct experiments on ResNet34 (He et al., 2016) using the attack approach Inverting Gradients (IG) (Geiping et al., 2020), see Figure 5.

## 7 Discussion and conclusion

During DP training, it is possible to save the perturbed gradients and accordingly find the coordinates whose gradients were small and possibly continue being insignificant in subsequent iterations. One might therefore expect better performance from a ranked sparsification algorithm, which for example ranks the mean of the perturbed gradients of last epoch and sparsifies accordingly. However, we find that due to the dominant noise, such ranking is in fact random, while on the other hand the resulting random sparsification still improves the performance. Similarly, other methods that select unimportant coordinates more precisely, e.g. DP selection or methods with public datasets, also cannot fully eliminate randomness. However, such random sparsification may still be beneficial for DP-SGD, which was not realized previously. More discussion are provided in Appendix I.

**Scope and Limitations**: In this work, we present a theoretical analysis of DP-SGD with random sparsification, uncovering a special trade-off between internal components in the convergence bound not observed in other popular SGD schemes. Our findings are supported by empirical evidence. By employing the proposed efficient and lightweight RS approach, we successfully enhance the performance of baseline models across various realistic settings. Additionally, the sparse gradients generated offer further advantages for addressing key challenges in private learning. However, our analyses do not guarantee that RS always converges faster

than non-sparsified approaches. In certain scenarios, such as DP transfer learning, RS may be less efficient, as discussed in the Appendix J. Developing a lower bound analysis of the convergence rate can also be interesting in terms of the uniqueness of the trade-off, which we leave for future work. Despite these limitations, we believe our insights on the intriguing interaction between RS and DP-SGD hold significant implications for the broader research community and can inspire further investigation of DP-SGD.

It is important to note that DP only reduces, rather than eliminates, the statistical dependency between its input and output. To ensure robust privacy protection, other techniques like cryptographic methods, including homomorphic encryption and secure multi-party computation, should be considered based on specific requirements and expectations for each individual application and policy context.

### Acknowledgments

This research received funding from the Flemish Government (AI Research Program) and the Research Foundation - Flanders (FWO) through project number G0G2921N.

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

# Appendix

In this appendix we provide full statements and proofs of our analyses (Appendix A-D). The pseudo code for noisy SGD, SGD with gradient clipping and DP-SGD with ranked sparsification are presented in Appendix E. Experimental results to verify the approximately symmetric property of the gradient deviation distribution $p(\xi_t)$ are demonstrated in Appendix F. More insights of gradual cooling are presented in Appendix G. Scaling rule for efficient hyperparameter tuning is discussed in Appendix H. More details of DP-SGD with ranked sparsifcation and connection between RS and selective sparsification or compression methods are given in Appendix I. Limitations of applying RS to DP transfer learning are discussed in Appendix J. Comparison between Poisson sampling and random shuffle is given in Appendix K.

**Contents**

# A Proof of applying random sparsification to noisy SGD

**Lemma A.1.** *$u, v \in \mathbb{R}^d$ are two arbitrary vectors, $m \in \{0, 1\}^d$ is a random mask with sparsification rate $r$. We have the following expectation, $\forall k \in \mathbb{Z}^+$,*

$$\mathbb{E}[\langle m^k \odot u, v \rangle] = (1 - r)\mathbb{E}[\langle u, v \rangle]. \tag{18}$$

*Proof.*

$$\mathbb{E}[\langle m^k \odot u, v \rangle] = \mathbb{E}[\langle m \odot u, v \rangle] \tag{19}$$

$$= \sum_i \mathbb{E}[m_i]\mathbb{E}[u_i v_i] \tag{20}$$

$$= (1 - r)\mathbb{E}[\langle u, v \rangle]. \tag{21}$$

$\square$

**Corollary A.2.** *Following the notations of Lemma A.1, we have the expectation below:*

$$\mathbb{E}[\langle m \odot u, m \odot v \rangle] = (1 - r)\mathbb{E}[\langle u, v \rangle]. \tag{22}$$

*Proof.*

$$\mathbb{E}[\langle m \odot u, m \odot v \rangle] = \mathbb{E}[\langle m^2 \odot u, v \rangle] \tag{23}$$

$$\overset{18}{=} (1 - r)\mathbb{E}[\langle u, v \rangle]. \tag{24}$$

$\square$

## A.1 Proof of Theorem 3.2

*Proof.* From the $G$-Lipschitz smoothness, and noting that Gaussian noise is added to the aggregated gradient and will be divided by batch-size $B$ before updating the model, we have:

$$\mathcal{L}_{t+1} \leq \mathcal{L}_t + \langle \nabla w_t, w_{t+1} - w_t \rangle + \frac{G}{2}\|w_{t+1} - w_t\|^2 \tag{25}$$

$$= \mathcal{L}_t - \gamma \langle \nabla w_t, m \odot (\nabla w_t + \frac{1}{B}(\sum_i \xi_{t,i} + \xi_\sigma))\rangle + \frac{G\gamma^2}{2}\|m \odot (\nabla w_t + \frac{1}{B}(\sum_i \xi_{t,i} + \xi_\sigma))\|^2. \tag{26}$$

Taking expectations on both sides and rearranging:

$$\gamma \mathbb{E}[\langle \nabla w_t, m \odot (\nabla w_t + \frac{1}{B}(\sum_i \xi_{t,i} + \xi_\sigma))\rangle]$$
$$\leq \mathbb{E}[\mathcal{L}_t - \mathcal{L}_{t+1}] + \frac{G\gamma^2}{2}\mathbb{E}[\|m \odot (\nabla w_t + \frac{1}{B}(\sum_i \xi_{t,i} + \xi_\sigma))\|^2], \tag{27}$$

from which we can obtain:

$$(1 - r)\gamma\|\nabla w_t\|^2 \overset{18}{\leq} \mathbb{E}[\mathcal{L}_t - \mathcal{L}_{t+1}] + \frac{(1 - r)G\gamma^2}{2}(\|\nabla w_t\|^2 + \mathbb{E}[\|\frac{1}{B}\sum_i \xi_{t,i}\|^2] + \frac{d\sigma^2}{B^2}) \tag{28}$$

$$\leq \mathbb{E}[\mathcal{L}_t - \mathcal{L}_{t+1}] + \frac{(1 - r)G\gamma^2}{2}(\|\nabla w_t\|^2 + \frac{\sigma_g^2}{B} + \frac{d\sigma^2}{B^2}). \tag{29}$$

We apply the same learning rate $\gamma = \frac{1}{G\sqrt{T}}$ as in Bernstein et al. (2018), with which we can derive better convergence under large noise. Combining $\|\nabla w_t\|^2$, we have:

$$\frac{1 - r}{2G\sqrt{T}}\|\nabla w_t\|^2 \leq (1 - r)(\frac{1}{G\sqrt{T}} - \frac{1}{2GT})\|\nabla w_t\|^2 \leq \mathbb{E}[\mathcal{L}_t - \mathcal{L}_{t+1}] + \frac{1 - r}{2GT}(\frac{\sigma_g^2}{B} + \frac{d\sigma^2}{B^2}). \tag{30}$$

Moving all the coefficients to the r.h.s.:

$$\|\nabla w_t\|^2 \leq \frac{2G\sqrt{T}}{1-r}\mathbb{E}[\mathcal{L}_t - \mathcal{L}_{t+1}] + \frac{1}{\sqrt{T}}(\frac{\sigma_g^2}{B} + \frac{d\sigma^2}{B^2}). \tag{31}$$

It is worth emphasizing that the factor $1-r$ on the r.h.s of Equation (30) is canceled out by the same factor from the l.h.s. Consider the average over $T$ steps:

$$\frac{1}{T}\sum_t \|\nabla w_t\|^2 \leq \frac{2G}{(1-r)\sqrt{T}}\sum_t \mathbb{E}[\mathcal{L}_t - \mathcal{L}_{t+1}] + \frac{1}{T\sqrt{T}}\sum_t(\frac{\sigma_g^2}{B} + \frac{d\sigma^2}{B^2}) \tag{32}$$

$$= \frac{2G}{(1-r)\sqrt{T}}\mathbb{E}[\mathcal{L}_1 - \mathcal{L}_{T+1}] + \frac{1}{\sqrt{T}}(\frac{\sigma_g^2}{B} + \frac{d\sigma^2}{B^2}) \tag{33}$$

$$\leq \frac{2G}{(1-r)\sqrt{T}}\mathbb{E}[\mathcal{L}_1] - \min_w \mathcal{L}(w) + \frac{1}{\sqrt{T}}(\frac{\sigma_g^2}{B} + \frac{d\sigma^2}{B^2}). \tag{34}$$

$\square$

## B    Proof of applying random sparsification SGD with gradient clipping

**Lemma B.1.** *$u \in \mathbb{R}^d$ is an arbitrary vector, $m \in \{0,1\}^d$ is a random mask with sparsification rate $r$. We have the following inequality:*

$$\mathbb{E}[\|m \odot u\|] \geq (1-r)\|u\|. \tag{35}$$

*Proof.*

$$\mathbb{E}[\|m \odot u\|] = \frac{1}{\|u\|}\mathbb{E}[\|m \odot u\|\|u\|] \tag{36}$$

$$\geq \frac{1}{\|u\|}\mathbb{E}[\|m \odot u\|^2] \tag{37}$$

$$\overset{22}{=} \frac{1}{\|u\|}(1-r)\mathbb{E}[\langle u, u\rangle] \tag{38}$$

$$= (1-r)\|u\|. \tag{39}$$

Equality holds when $r = 0$. $\square$

### B.1    Proof of Lemma 3.1

*Proof.* Following from the G-smoothness assumption, we have:

$$\mathcal{L}_{t+1} \leq \mathcal{L}_t + \langle \nabla w_t, w_{t+1} - w_t\rangle + \frac{G}{2}\|w_{t+1} - w_t\|^2 \tag{40}$$

$$= \mathcal{L}_t - \gamma\langle \nabla w_t, \bar{g}_t\rangle + \frac{G\gamma^2}{2}\|\bar{g}_t\|^2. \tag{41}$$

Taking expectations on both sides and rearranging, we have:

$$\mathbb{E}[\langle \nabla w_t, \bar{g}_t\rangle] \leq \frac{1}{\gamma}\mathbb{E}[\mathcal{L}_t - \mathcal{L}_{t+1}] + \frac{G\gamma}{2}\mathbb{E}[\|\bar{g}_t\|^2] \tag{42}$$

$$\leq \frac{1}{\gamma}\mathbb{E}[\mathcal{L}_t - \mathcal{L}_{t+1}] + \gamma\frac{GC^2}{2}, \tag{43}$$

$\square$

## B.2 Proof of Equation (9) and Equation (10)

Similar to Lemma 3.1, consider gradient sparsification then clipping, we can obtain:

$$\mathbb{E}[\langle \nabla w_t, \hat{g}_t \rangle] \leq \frac{1}{\gamma}\mathbb{E}[\mathcal{L}_t - \mathcal{L}_{t+1}] + \gamma\frac{GC^2}{2}. \tag{44}$$

Now focusing on the l.h.s. we have:

$$\mathbb{E}[\langle \nabla w_t, \hat{g}_t \rangle] = \langle \nabla w_t, \mathbb{E}[\hat{g}_t] \rangle \tag{45}$$

$$= \langle \nabla w_t, \frac{1}{B}\sum_i \mathbb{E}[m \odot g_{t,i} \cdot \min(1, \frac{C}{\|m \odot g_{t,i}\|})] \rangle \tag{46}$$

$$= \mathbb{E}[\langle \nabla w_t, m \odot g_{t,i} \cdot \min(1, \frac{C}{\|m \odot g_{t,i}\|}) \rangle] \tag{47}$$

$$= \mathbb{E}[\langle m \odot \nabla w_t, m \odot g_{t,i} \cdot \min(1, \frac{C}{\|m \odot g_{t,i}\|}) \rangle] \tag{48}$$

$$= \mathbb{E}_m\left[\mathbb{E}_{\xi_t}[\langle \nabla w'_t, g'_{t,i} \cdot \min(1, \frac{C}{\|g'_{t,i}\|}) \rangle]\right] \tag{49}$$

$$= \mathbb{E}_m\left[\mathbb{E}_{\xi'_t \sim \tilde{p}'_t}[\langle \nabla w'_t, g'_{t,i} \cdot \min(1, \frac{C}{\|g'_{t,i}\|}) \rangle]\right] + \mathbb{E}_m[b'_t], \tag{50}$$

where we define $\nabla w'_t := m \odot \nabla w_t$, $\xi'_t := m \odot \xi_t$ with corresponding true distribution $p'_t$ and proxy $\tilde{p}'_t$ which are projected from $p_t$ and $\tilde{p}_t$, $b'_t := \mathbb{E}_{\xi'_t \sim p'_t}[\langle \nabla w', \hat{g}_t \rangle] - \mathbb{E}_{\xi'_t \sim \tilde{p}'_t}[\langle \nabla w', \hat{g}_t \rangle]$. Equation (49) holds because $\xi_t$ and $m$ are independent. Since the projection of a symmetric distribution to a subspace is symmetric, i.e. $\tilde{p}'_t(\xi'_t) = \tilde{p}'_t(-\xi'_t)$. We can adapt Theorem 3.3 for the first term:

$$\mathbb{E}_m\left[\mathbb{E}_{\xi'_t \sim \tilde{p}'_t}[\langle \nabla w'_t, g'_{t,i} \cdot \min(1, \frac{C}{\|g'_{t,i}\|}) \rangle]\right] \geq \mathbb{E}_m[P_{\xi'_t \sim \tilde{p}'_t}(\|\xi'_t\| < 3C/4)h(\nabla w'_t)\|\nabla w'_t\|]. \tag{51}$$

## B.3 Proof of Lemma 3.3

*Proof.* Consider the following two cases:

*case 1*: $\|\nabla w_t\| \leq C/4$, then $P_m(\|\nabla w'_t\| \leq C/4) = 1$,

$$\mathbb{E}_m[P_{\xi'_t \sim \tilde{p}'_t}(\|\xi'_t\| < 3C/4)h(\nabla w'_t)\|\nabla w'_t\|] \geq P_{\xi_t \sim \tilde{p}_t}(\|\xi_t\| < 3C/4)\mathbb{E}_m[h(\nabla w'_t)\|\nabla w'_t\|] \tag{52}$$

$$= P_{\xi_t \sim \tilde{p}_t}(\|\xi_t\| < 3C/4)\mathbb{E}_m[\|\nabla w'_t\|^2] \tag{53}$$

$$\overset{22}{=} P_{\xi_t \sim \tilde{p}_t}(\|\xi_t\| < 3C/4)(1-r)\|\nabla w_t\|^2 \tag{54}$$

$$= (1-r)P_{\xi_t \sim \tilde{p}_t}(\|\xi_t\| < 3C/4)h(\nabla w_t)\|\nabla w_t\|. \tag{55}$$

*case 2*: $\|\nabla w_t\| > C/4$, consider two events $A_1$: $\|\nabla w'_t\| \leq C/4$; $A_2$: $\|\nabla w'_t\| > C/4$ then we have:

$$\mathbb{E}_m[P_{\xi'_t \sim \tilde{p}'_t}(\|\xi'_t\| < 3C/4)h(\nabla w'_t)\|\nabla w'_t\|] \geq P_{\xi_t \sim \tilde{p}_t}(\|\xi_t\| < 3C/4)\mathbb{E}_m[h(\nabla w'_t)\|\nabla w'_t\|] \tag{56}$$

$$\overset{54}{\geq} P_{\xi_t \sim \tilde{p}_t}(\|\xi_t\| < 3C/4)(P(A_1)(1-r)\|\nabla w_t\|^2 + \tag{57}$$
$$P(A_2)C/4 \cdot \mathbb{E}_m[\|\nabla w'_t\|])$$

$$\overset{35}{\geq} P_{\xi_t \sim \tilde{p}_t}(\|\xi_t\| < 3C/4)(P(A_1)(1-r)\|\nabla w_t\|^2 + \tag{58}$$
$$P(A_2)\frac{C}{4}(1-r)\|\nabla w_t\|)$$

$$\geq P_{\xi_t \sim \tilde{p}_t}(\|\xi_t\| < 3C/4)(1-r)\frac{C}{4}\|\nabla w_t\| \tag{59}$$

$$= (1-r)P_{\xi_t \sim \tilde{p}_t}(\|\xi_t\| < 3C/4)h(\nabla w_t)\|\nabla w_t\|, \tag{60}$$

from which we conclude that:

$$\mathbb{E}_m[P_{\xi'_t \sim \tilde{p}'_t}(\|\xi'_t\| < 3C/4)h(\nabla w'_t)\|\nabla w'_t\|] \geq (1-r)P_{\xi_t \sim \tilde{p}_t}(\|\xi_t\| < 3C/4)h(\nabla w_t)\|\nabla w_t\|. \tag{61}$$

The inequality above implies that $\exists \kappa_t \geq 1 - r$ such that:

$$\mathbb{E}_m[P_{\xi'_t \sim \tilde{p}'_t}(\|\xi'_t\| < 3C/4)h(\nabla w'_t)\|\nabla w'_t\|] = \kappa_t P_{\xi_t \sim \tilde{p}_t}(\|\xi_t\| < 3C/4)h(\nabla w_t)\|\nabla w_t\|. \tag{62}$$

We note that $\kappa_t = 1$ if $r = 0$ and under natural assumption, e.g. $P_{\xi'_t \sim \tilde{p}'_t}(\|\xi'_t\| < 3C/4) > P_{\xi_t \sim \tilde{p}_t}(\|\xi_t\| < 3C/4)$, we have $\kappa_t > 1 - r$ if $r > 0$.

Now we form the lower bound for $\kappa_t$ using the condition $P_{\xi_t \sim \tilde{p}_t}(\|\xi_t\| < 3C/4) \geq \sqrt{1-r}$. We emphasize that as the network converges during training, the following property will be satisfied with the true probability distribution $p_t$:

$$P_{\xi_t \sim p_t}(\|\xi_t\| < 3C/4) \geq \sqrt{1-r}. \tag{63}$$

So the condition of choosing $\tilde{p}_t$ will not lead to significant bias term $b_t$. Based on the condition, we have:

$$P_{\xi_t \sim \tilde{p}_t}(\|\xi_t\| < 3C/4)h(\nabla w)\|\nabla w\| \geq \sqrt{1-r}h(\nabla w)\|\nabla w\|. \tag{64}$$

Consider two cases:

*case 1*: $\|\nabla w\| \geq C/4$, r.h.s. of Equation (64) has:

$$\sqrt{1-r}h(\nabla w)\|\nabla w\| \geq h(\nabla w)((1-r)\|\nabla w\|^2)^{1/2} \tag{65}$$

$$\overset{22}{=} h(\nabla w)(\mathbb{E}_m[\|\nabla w'\|^2])^{1/2} \tag{66}$$

$$\geq h(\nabla w)\mathbb{E}_m[\|\nabla w'\|] \tag{67}$$

$$= \mathbb{E}_m[h(\nabla w)\|\nabla w'\|] \tag{68}$$

$$\geq \mathbb{E}_m[h(\nabla w')\|\nabla w'\|], \tag{69}$$

where Equation (67) is taken according to Jensens' inequality.

*case 2*: $\|\nabla w\| \leq C/4$, so we have $\|\nabla w'\| \leq C/4$, then:

$$\sqrt{1-r}h(\nabla w)\|\nabla w\| = \sqrt{1-r}\|\nabla w\|^2 \tag{70}$$

$$\geq (1-r)\|\nabla w\|^2 \tag{71}$$

$$\overset{22}{=} \mathbb{E}_m[\|\nabla w'\|^2] \tag{72}$$

$$= \mathbb{E}_m[h(\nabla w')\|\nabla w'\|]. \tag{73}$$

Combing two cases, we can obtain the following property:

$$P_{\xi_t \sim \tilde{p}_t}(\|\xi_t\| < 3C/4)h(\nabla w)\|\nabla w\| \geq \sqrt{1-r}h(\nabla w)\|\nabla w\| \tag{74}$$

$$\geq \mathbb{E}_m[h(\nabla w')\|\nabla w'\|] \tag{75}$$

$$\geq \mathbb{E}_m[P_{\xi'_t \sim \tilde{p}'_t}(\|\xi'_t\| < 3C/4)h(\nabla w')\|\nabla w'\|]. \tag{76}$$

It is worth noting that the above inequality can also be obtained under other conditions, the one we provided, i.e. $P_{\xi_t \sim \tilde{p}_t}(\|\xi_t\| < 3C/4) \geq \sqrt{1-r}$, is a sufficient but not necessary condition, which will be definitely satisfied as the network converges, i.e. $\|\nabla w\| \to 0$, $\|\xi_t\| \to 0$.

Combining Equation (76) with Equation (61), we obtain: $\exists \kappa_t \in (1-r, 1)$,

$$\mathbb{E}_m[P_{\xi'_t \sim \tilde{p}'_t}(\|\xi'_t\| < 3C/4)h(\nabla w'_t)\|\nabla w'_t\|] = \kappa_t P_{\xi_t \sim \tilde{p}_t}(\|\xi_t\| < 3C/4)h(\nabla w_t)\|\nabla w_t\|. \tag{77}$$

$\kappa_t$ takes 1 when $r = 0$. $\qquad\square$

## C  Proof of applying random sparsification to DP-SGD

### C.1  Proof of Lemma 3.4

*Proof.* Following from the smoothness assumption, we have:

$$\mathcal{L}_{t+1} \leq \mathcal{L}_t + \langle \nabla w_t, w_{t+1} - w_t \rangle + \frac{G}{2} \|w_{t+1} - w_t\|^2 \tag{78}$$

$$= \mathcal{L}_t - \gamma \langle \nabla w_t, \hat{g}_t + m \odot \xi_{DP} \rangle + \frac{G\gamma^2}{2} \|\hat{g}_t + m \odot \xi_{DP}\|^2 \tag{79}$$

$$= \mathcal{L}_t - \gamma \langle \nabla w_t, \hat{g}_t \rangle - \gamma \langle \nabla w_t, m \odot \xi_{DP} \rangle + \frac{G\gamma^2}{2} \|\hat{g}_t + m \odot \xi_{DP}\|^2, \tag{80}$$

taking expectations on both sides and rearranging, we have:

$$\mathbb{E}[\langle \nabla w_t, \hat{g}_t \rangle] \leq \frac{1}{\gamma} \mathbb{E}[\mathcal{L}_t - \mathcal{L}_{t+1}] - \mathbb{E}[\langle \nabla w_t, m \odot \xi_{DP} \rangle] + \frac{G\gamma}{2} \mathbb{E}[\|\hat{g}_t + m \odot \xi_{DP}\|^2] \tag{81}$$

$$= \frac{1}{\gamma} \mathbb{E}[\mathcal{L}_t - \mathcal{L}_{t+1}] - 0 + \frac{G\gamma}{2} (\mathbb{E}[\|\hat{g}_t\|^2] + \mathbb{E}[\|m \odot \xi_{DP}\|^2] - 0) \tag{82}$$

$$\overset{22}{=} \frac{1}{\gamma} \mathbb{E}[\mathcal{L}_t - \mathcal{L}_{t+1}] + \frac{G\gamma}{2} (\mathbb{E}[\|\hat{g}_t\|^2] + (1-r)\frac{C^2 \sigma^2 d}{B^2}) \tag{83}$$

$$\leq \frac{1}{\gamma} \mathbb{E}[\mathcal{L}_t - \mathcal{L}_{t+1}] + \frac{G\gamma}{2} (C^2 + (1-r)\frac{C^2 \sigma^2 d}{B^2}). \tag{84}$$

$\square$

### C.2  Proof of Theorem 3.5

*Proof.* Combine Equation (9), 10 and Lemma 3.3:

$$\kappa_t P_{\xi_t \sim \tilde{p}_t}(\|\xi_t\| < 3C/4) h(\nabla w_t) \|\nabla w_t\| + \mathbb{E}_m[b'_t] \leq \mathbb{E}[\langle \nabla w_t, \hat{g}_t \rangle]. \tag{85}$$

Substitute Lemmma 3.4 and rearrange:

$$\kappa_t P_{\xi_t \sim \tilde{p}_t}(\|\xi_t\| < 3C/4) h(\nabla w_t) \|\nabla w_t\| \leq \mathbb{E}[\langle \nabla w_t, \hat{g}_t \rangle] - \mathbb{E}_m[b_t], \tag{86}$$

$$\kappa_t P_{\xi_t \sim \tilde{p}_t}(\|\xi_t\| < 3C/4) h(\nabla w_t) \|\nabla w_t\| \leq \frac{1}{\gamma} \mathbb{E}[\mathcal{L}_t - \mathcal{L}_{t+1}] + \gamma \frac{GC^2}{2} + (1-r)\gamma \Delta_\sigma - \mathbb{E}_m[b'_t], \tag{87}$$

$$P_{\xi_t \sim \tilde{p}_t}(\|\xi_t\| < 3C/4) h(\nabla w_t) \|\nabla w_t\| \leq \frac{1}{\kappa_t} (\frac{\mathbb{E}[\mathcal{L}_t - \mathcal{L}_{t+1}]}{\gamma} + \gamma \frac{GC^2}{2} - \mathbb{E}_m[b'_t]) + \frac{1-r}{\kappa_t} \gamma \Delta_\sigma. \tag{88}$$

Since $\kappa_{1:T} \in (1-r, 1)$, then $\exists \kappa \in (1-r, 1)$, such that consider all T iterations, we have:

$$\frac{1}{T} \sum_{t=1}^{T} P_{\xi_t \sim \tilde{p}_t}(\|\xi_t\| < 3C/4) h(\nabla w_t) \|\nabla w_t\| \leq \frac{1}{\kappa} (\frac{\Delta_\mathcal{L}}{\gamma T} + \gamma \frac{GC^2}{2} - \frac{1}{T} \sum_{t=1}^{T} \mathbb{E}_m[b'_t]) + \frac{1-r}{\kappa} \gamma \Delta_\sigma, \tag{89}$$

$\square$

### C.3  Convergence bound with privacy budget variables

**Theorem C.1** (Abadi et al. (2016b)). *There exist constants u and v so that give sampling probability $q = B/N$ and the number of steps T, for any $\varepsilon < vq^2 T$, DP-SGD is $(\varepsilon, \delta)$-differentially private for any $\delta > 0$ if we choose:*

$$\sigma \geq u \frac{q\sqrt{T \log(1/\delta)}}{\varepsilon}. \tag{90}$$

So to achieve $(\varepsilon, \delta)$-differential privacy, we choose $\sigma = u\frac{q\sqrt{T\log(1/\delta)}}{\varepsilon}$ and note that for DP-SGD, large batch is preferred in practice, the following corollary proves that the upper bound of true gradient norm $\nabla w_t$ decreases w.r.t. iterations $T$:

**Corollary C.1.1.** *Follow Theorem 3.5, consider the overall $T$ iterations with learning rate $\gamma = \frac{1}{G\sqrt{T}}$, batch-size $B = \sqrt{T}$, privacy budget $(\varepsilon, \delta)$, we have:*

$$\frac{1}{T}\sum_{t=1}^{T} P_{\xi_t \sim \tilde{p}_t}(\|\xi_t\| < 3C/4)h(\nabla w_t)\|\nabla w_t\| \leq \frac{1}{\kappa}(\frac{\Delta_{\mathcal{L}}G}{\sqrt{T}} + \frac{C^2}{2\sqrt{T}} - \frac{1}{T}\sum_{t=1}^{T}\mathbb{E}_m[b_t']) + \frac{1-r}{\kappa}\Delta_{\varepsilon}, \tag{91}$$

*where we define $\Delta_{\varepsilon} := \frac{dC^2u^2q^2\log(1/\delta)}{2\sqrt{T}\varepsilon^2}$.*

*Proof.*

$$\frac{1}{T}\sum_{t=1}^{T} P_{\xi_t \sim \tilde{p}_t}(\|\xi_t\| < 3C/4)h(\nabla w_t)\|\nabla w_t\| \tag{92}$$

$$\leq \frac{1}{\kappa}(\frac{\Delta_{\mathcal{L}}}{\gamma T} + \gamma\frac{GC^2}{2} - \frac{1}{T}\sum_{t=1}^{T}\mathbb{E}_m[b_t']) + \frac{1-r}{\kappa}\gamma\Delta_{\sigma} \tag{93}$$

$$= \frac{1}{\kappa}(\frac{\Delta_{\mathcal{L}}G}{\sqrt{T}} + \frac{C^2}{2\sqrt{T}} - \frac{1}{T}\sum_{t=1}^{T}\mathbb{E}_m[b_t']) + \frac{1-r}{\kappa}\gamma\Delta_{\sigma}. \tag{94}$$

Focusing on $\gamma\Delta_{\sigma}$:

$$\gamma\Delta_{\sigma} = \frac{\gamma C^2 dG}{2B^2} \cdot \frac{u^2q^2T\log(1/\delta)}{\varepsilon^2} \tag{95}$$

$$= \frac{\gamma C^2 dG}{2} \cdot \frac{u^2q^2\log(1/\delta)}{\varepsilon^2} \tag{96}$$

$$= \frac{dC^2u^2q^2\log(1/\delta)}{2\sqrt{T}\varepsilon^2}, \tag{97}$$

Plugging Equation (97) into Equation (94):

$$\frac{1}{T}\sum_{t=1}^{T} P_{\xi_t \sim \tilde{p}_t}(\|\xi_t\| < 3C/4)h(\nabla w_t)\|\nabla w_t\| \leq \frac{1}{\kappa}(\frac{\Delta_{\mathcal{L}}G}{\sqrt{T}} + \frac{C^2}{2\sqrt{T}} - \frac{1}{T}\sum_{t=1}^{T}\mathbb{E}_m[b_t']) + \frac{1-r}{\kappa}\Delta_{\varepsilon}. \tag{98}$$

$\square$

# D Proof of privacy against gradient reconstruction attacks

## D.1 Proof of Theorem 5.1

*Proof.* Analogous to $Kx = g$ (Zhu & Blaschko, 2021, Equation (15)), we have the following system of linear equations:

$$m \odot kx = m \odot g. \tag{99}$$

Assume that $m \odot k \neq 0$ (otherwise the reconstruction will fail, i.e. arbitrary reconstruction error), $x$ can be estimated through least squares:

$$\hat{x} = \left((m \odot k)^{\top}(m \odot k)\right)^{-1}(m \odot k)^{\top}(m \odot (g + \xi_{\sigma})). \tag{100}$$

The squared error of reconstruction can be expressed as:

$$\|x - \hat{x}\|^2 = \|x - \left((m \odot k)^\top (m \odot k)\right)^{-1} (m \odot k)^\top (m \odot (g + \xi_\sigma))\|^2 \tag{101}$$

$$= \|x - \left((m \odot k)^\top (m \odot k)\right)^{-1} (m \odot k)^\top (g + \xi_\sigma)\|^2 \tag{102}$$

$$= \|\left((m \odot k)^\top (m \odot k)\right)^{-1} (m \odot k)^\top \xi_\sigma\|^2. \tag{103}$$

Take expectations on both sides:

$$\mathbb{E}[\|x - \hat{x}\|^2] = \mathbb{E}[\|\left((m \odot k)^\top (m \odot k)\right)^{-1} (m \odot k)^\top \xi_\sigma\|^2 \tag{104}$$

$$= \mathbb{E}[\xi_\sigma^\top \frac{m \odot k}{\|m \odot k\|^2} \frac{(m \odot k)^\top}{\|m \odot k\|^2} \xi] \tag{105}$$

$$= \mathbb{E}[\mathrm{Tr}\{\xi_\sigma \xi_\sigma^\top \frac{m \odot k}{\|m \odot k\|^2} \frac{(m \odot k)^\top}{\|m \odot k\|^2}\}] \tag{106}$$

$$= \mathrm{Tr}\{\mathbb{E}[\xi_\sigma \xi_\sigma^\top]\mathbb{E}[\frac{m \odot k}{\|m \odot k\|^2} \frac{(m \odot k)^\top}{\|m \odot k\|^2}]\} \tag{107}$$

$$= \mathrm{Tr}\{\sigma^2 \boldsymbol{I}_d \mathbb{E}[\frac{m \odot k}{\|m \odot k\|^2} \frac{(m \odot k)^\top}{\|m \odot k\|^2}]\} \tag{108}$$

$$= \sigma^2 \mathbb{E}[\mathrm{Tr}\{\frac{m \odot k}{\|m \odot k\|^2} \frac{(m \odot k)^\top}{\|m \odot k\|^2}\}] \tag{109}$$

$$= \sigma^2 \mathbb{E}[\frac{1}{\|m \odot k\|^2}]. \tag{110}$$

We see that as more gradients are zeroed out, the expected squared error increases. In particular, according to Jensen's inequality and the convexity of $1/\|m \odot k\|^2$:

$$\mathbb{E}[\|x - \hat{x}\|^2] \geq \frac{\sigma^2}{\mathbb{E}[\|m \odot k\|^2]} \tag{111}$$

$$\overset{22}{=} \frac{\sigma^2}{(1 - r)\|k\|^2}. \tag{112}$$

The lower bound of expected squared error increases monotonically with increasing sparsification rate $r$. □

# E   Additional algorithms

We provide pseudo codes for Noisy SGD with RS (Algorithm 2), SGD with gradient clipping and RS (Algorithm 3), and DP-SGD with ranked sparsification (Algorithm 4).

---

**Algorithm 2** Noisy SGD with Random Sparsification

**Input:** Initial parameters $w_0$; Epochs $E$; Batch size $B$; Sparsification rate: $r^*$; Momentum: $\mu$; Learning rate $\gamma$; Noise multiplier $\sigma$.

1: **for** $e = 0$ to $E - 1$ **do**
2:     $\triangleright$ *Gradual cooling*     $\triangleleft$
3:     $r(e) = r^* \cdot \frac{e}{E-1}$;
4:     $\triangleright$ *Generate a random mask every epoch*     $\triangleleft$
5:     $m \in \{0,1\}^d$, s.t. $\|m\|_1 = d \cdot (1 - r(e))$;
6:     **for** $t = 0$ to $T - 1$ **do**
7:        $\triangleright$ *Compute aggregated gradients*     $\triangleleft$
8:        $g_t = \sum_i \nabla \ell(w_t, x_i)$;
9:        $\triangleright$ *Sparsify gradient*     $\triangleleft$
10:       $g'_t = m \odot g_t$;
11:       $\triangleright$ *Add sparsified noise*     $\triangleleft$
12:       $\hat{g}_t = \frac{1}{B}(g'_t + m \odot \mathcal{N}(0, \sigma^2 \boldsymbol{I}_d))$;
13:       $\triangleright$ *Update parameters*     $\triangleleft$
14:       $v_{t+1} = \mu \cdot v_t + \hat{g}_t, w_{t+1} = w_t - \gamma v_{t+1}$;

---

**Algorithm 3** SGD with Gradient clipping and Random Sparsification

**Input:** Initial parameters $w_0$; Epochs $E$; Batch size $B$; Sparsification rate: $r^*$; Clipping bound: $C$; Momentum: $\mu$; Learning rate $\gamma$.

1: **for** $e = 0$ to $E - 1$ **do**
2:     $\triangleright$ *Gradual cooling*     $\triangleleft$
3:     $r(e) = r^* \cdot \frac{e}{E-1}$;
4:     $\triangleright$ *Generate a random mask every epoch*     $\triangleleft$
5:     $m \in \{0,1\}^d$, s.t. $\|m\|_1 = d \cdot (1 - r(e))$;
6:     **for** $t = 0$ to $T - 1$ **do**
7:        $\triangleright$ *For each $x_i$ in the minibatch of size $B$*     $\triangleleft$
8:        $g_{t,i} = \nabla \ell(w_t, x_i)$;
9:        $\triangleright$ *Sparsify gradient*     $\triangleleft$
10:       $g'_{t,i} = m \odot g_{t,i}$;
11:       $\triangleright$ *Clip each individual gradient*     $\triangleleft$
12:       $\hat{g}_t = \frac{1}{B} \sum_i g'_{t,i} \cdot \min(1, C/\|g'_{t,i}\|)$;
13:       $\triangleright$ *Update parameters*     $\triangleleft$
14:       $v_{t+1} = \mu \cdot v_t + \hat{g}_t, w_{t+1} = w_t - \gamma v_{t+1}$;

---

---

**Algorithm 4** DP-SGD with Ranked Sparsification

---

**Input:** Initial parameters $w_0$; Epochs $E$; Batch size $B$; Sparsification rate: $r^*$; Clipping bound: $C$; Momentum: $\mu$; Learning rate $\gamma$; Noise multiplier $\sigma$.

1: **for** $e = 0$ to $E - 1$ **do**
2:      $z_e = \{0\}^d$;
3:      ▷ *Gradual Cooling*                                                    ◁
4:      $r(e) = r^* \cdot \frac{e}{E-1}$;
5:      ▷ *Generate a ranked mask every epoch*                      ◁
6:      **if** $e$ is 0 **then**
7:          $m = \{1\}^d$;
8:      **else**
9:          Sort the indices $[1, ..., d]$ with respect to the magnitude of aggregated gradient of the last epoch $|z^{e-1}|$ in ascending order then set the first $d \cdot r(e)$ positions in mask $m$ to 0 and the rest to 1;
10:      **for** $t = 0$ to $T - 1$ **do**
11:          ▷ *For each $x_i$ in the minibatch of size $B$*                  ◁
12:          $g_{t,i} = \nabla \ell(w_t, x_i)$;
13:          ▷ *Sparsify gradient*                                           ◁
14:          $g'_{t,i} = m \odot g_{t,i}$;
15:          ▷ *Clip each individual gradient*                         ◁
16:          $\hat{g}_{t,i} = g'_{t,i} \cdot \min(1, C/\|g'_{t,i}\|)$;
17:          ▷ *Add sparsified noise*                                    ◁
18:          $\tilde{g}_t = \frac{1}{B}(\sum_i \hat{g}_{t,i} + m \odot \mathcal{N}(0, C^2 \sigma^2 \boldsymbol{I}_d))$;
19:          ▷ *Update parameters*                                    ◁
20:          $v_{t+1} = \mu \cdot v_t + \tilde{g}_t$, $w_{t+1} = w_t - \gamma v_{t+1}$;
21:          ▷ *Save perturbed gradients*                              ◁
22:          $z^e = z^e + \tilde{g}_t + \frac{1}{B}(1 - m) \odot \mathcal{N}(0, C^2 \sigma^2 \boldsymbol{I}_d)$;

---

## F  Approximately symmetric distribution of the gradient deviation

As the gradient deviation distribution $p(\xi_t)$ is high-dimensional, demonstrating and verifying its symmetricity is in general intractable. We therefore adopt random projection onto a 2D for the visualization, which is also adopted by Chen et al. (2020). Our experiments reproduce their observation that the gradient deviation distribution approximates symmetry during training with DP-SGD (see Figure 6 and 7). We verify that this property is also valid for DP-SGD with RS (see Figure 8 and 9). We have repeated the experiments 10 times, all results are qualitatively the same as presented here.

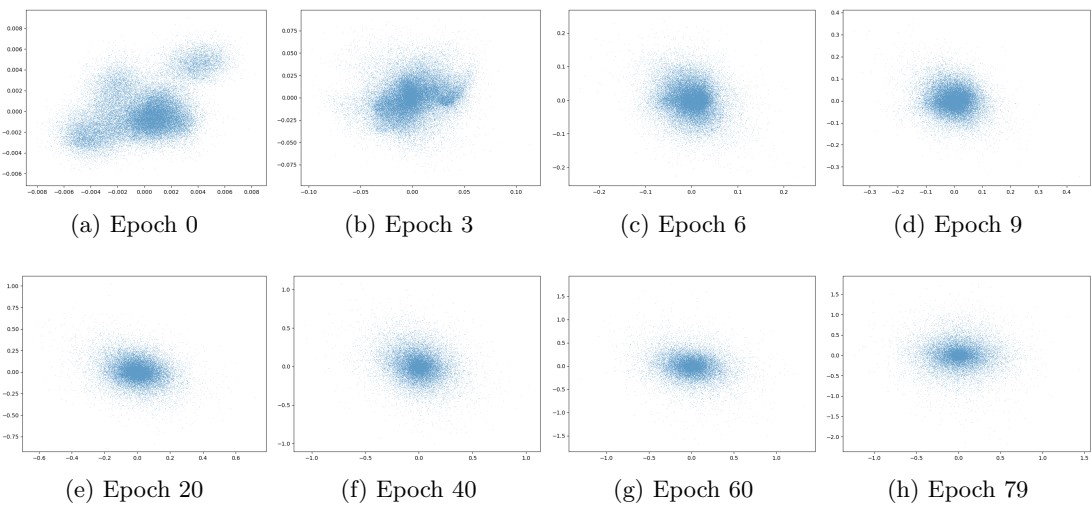

Figure 6: Gradient deviation distribution during training, projected onto a 2D using a random matrix. Network is DP-CNN, method is DP-SGD, dataset is CIFAR10.

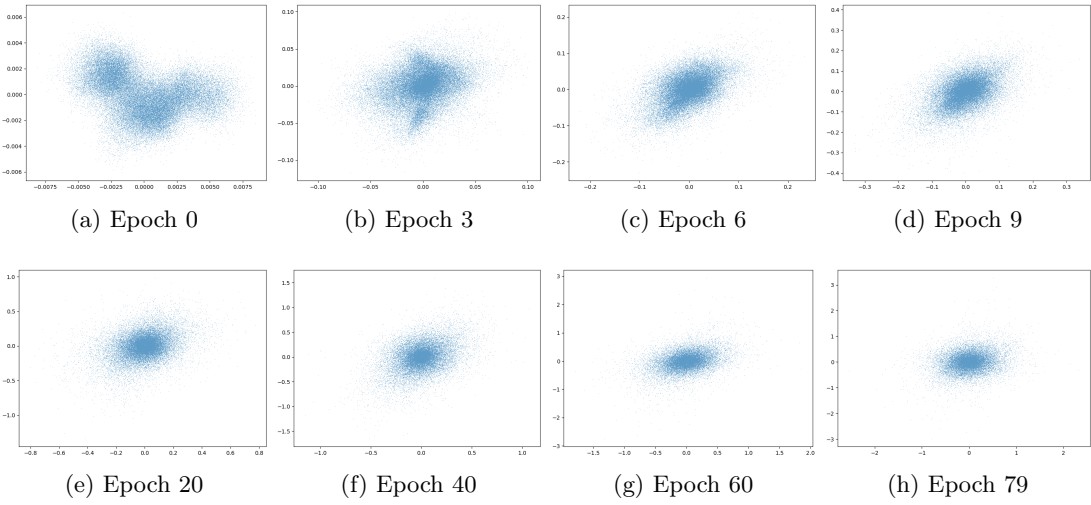

Figure 7: Gradient deviation distribution during training, projected onto a 2D using a random matrix. Network is DP-CNN, method is DP-SGD, dataset is CIFAR10.

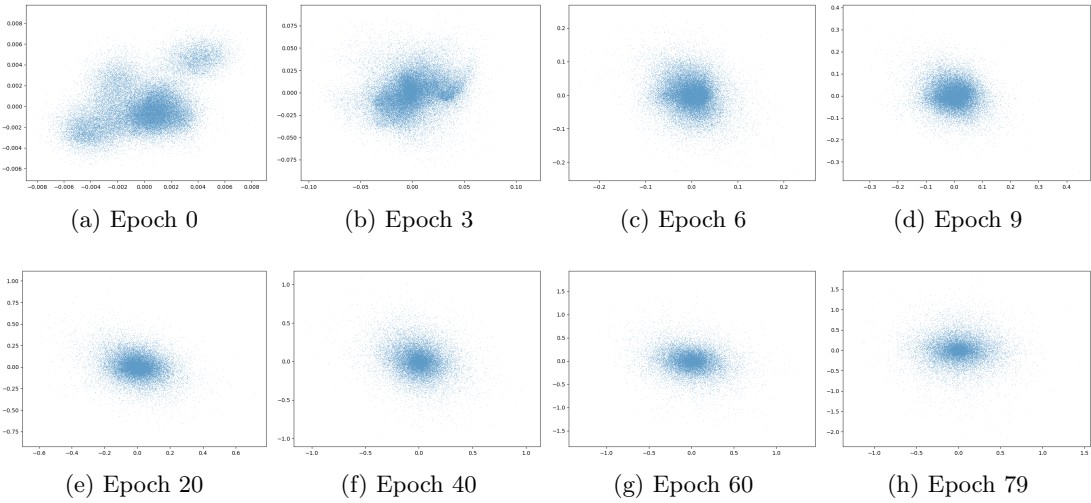

(a) Epoch 0     (b) Epoch 3     (c) Epoch 6     (d) Epoch 9

(e) Epoch 20     (f) Epoch 40     (g) Epoch 60     (h) Epoch 79

Figure 8: Gradient deviation distribution during training, projected onto a 2D using a random matrix. Network is DP-CNN, method is DP-SGD with RS, dataset is CIFAR10.

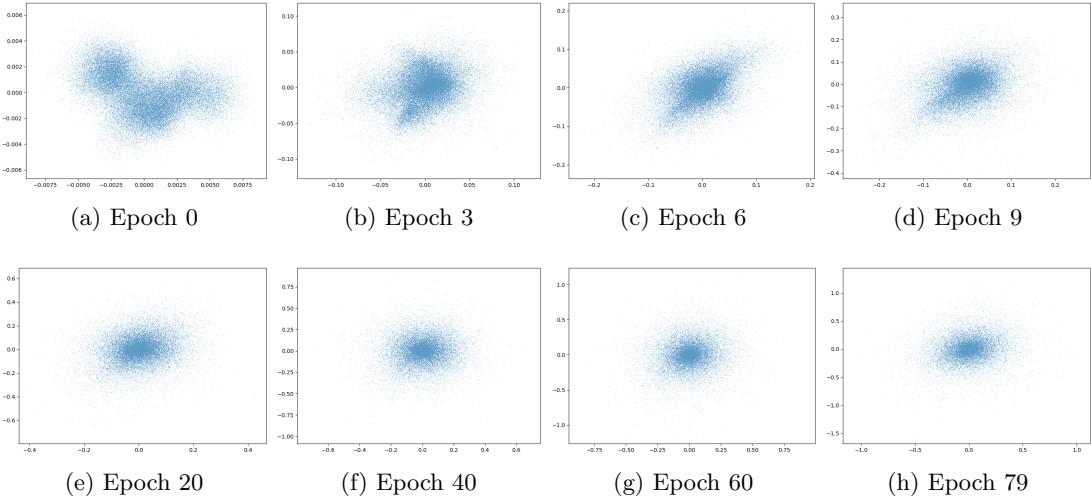

(a) Epoch 0     (b) Epoch 3     (c) Epoch 6     (d) Epoch 9

(e) Epoch 20     (f) Epoch 40     (g) Epoch 60     (h) Epoch 79

Figure 9: Gradient deviation distribution during training, projected onto a 2D using a random matrix. Network is DP-CNN, method is DP-SGD with RS, dataset is CIFAR10.

## G   Further discussion of gradual cooling

For the convenience we restate Equation (14) here, and for the clarity we define $\Delta_C := \frac{\Delta_{\mathcal{L}}}{\gamma T} + \gamma \frac{GC^2}{2} - \frac{1}{T}\sum_{t=1}^{T}\mathbb{E}_m[b'_t]$ to denote the terms that are enlarged by RS:

$$\frac{1}{T}\sum_{t=1}^{T} P_{\xi_t \sim \tilde{p}_t}(\|\xi_t\| < 3C/4)h(\nabla w_t)\|\nabla w_t\| \leq \frac{1}{\kappa}\Delta_C + \frac{1-r}{\kappa}\gamma\Delta_\sigma. \tag{113}$$

We see $\Delta_\sigma = C^2\sigma^2 dG/2B^2$ can be computed given the constants, assume $\Delta_C$ is also given, to deduce the optimal sparsification rate, we need to relate $\kappa$ to $r$. However, beyond giving a range $\kappa \in (1-r, 1)$, it is not possible to represent $\kappa$ by $r$ without supposing an additional assumption over the shape of the distribution of the gradient deviation, which is difficult to make. We therefore start from the empirical evidence in Figure 3.

Recall that the factor introduced by RS for noisy SGD is $1/(1 - r)$ (see Theorem 3.2) and for SGD with gradient clipping is $1/k$ (see Theorem 3.4). From Figures 3a, 3b, 3c, we see that as $r$ increases SGD with gradient clipping also obviously converges slower, which implies that $1/k$ also grows with $r$. From Figure 3d, 3e, 3f, we find that the performance of noisy SGD tends to drop more dramatically in a high sparsification regime, like a parabolic shape, which implies $1/k$ is comparatively more stable than $1/(1 - r)$ as $r$ changes.

Based on these observations from Figure 2, we suppose that $\kappa = \sqrt{1 - r}$, which complies with the theoretical analysis that $\kappa = 1$ when $r = 0$ and $\kappa \in (1 - r, 1)$. Using this relation we can express the convergence bound as a function of sparsification rate $r$:

$$U(r) := \frac{1}{\sqrt{1 - r}} \Delta_C + \sqrt{1 - r} \Delta_\sigma. \tag{114}$$

To find the optimal $r^*$, we can simply compute the derivative of $U$:

$$U'(r) = 1/2(1 - r)^{-3/2} \Delta_C - 1/2(1 - r)^{-1/2} \Delta_\sigma \tag{115}$$

Consider the first order condition, we have:

$$r^* = 1 - \Delta_C / \Delta_\sigma. \tag{116}$$

So when $\Delta_C > \Delta_\sigma$, i.e. the noise term is insignificant in the convergence bound, we have $r^* = 0$ is optimal. While when $\Delta_\sigma > \Delta_C$, we have $r^* = 1 - \Delta_C / \Delta_\sigma$ is optimal, and as $\Delta_\sigma$ becomes dominant, higher sparsification rate is preferred. We see this conclusion matches our empirical evidence in Figure 2.

Although in practice $\Delta_C$ is unknown and $r^*$ therefore cannot be precisely estimated, the observation and conclusion above reason and support the gradual cooling. At early training stages, the network converges fast: $\mathbb{E}[\mathcal{L}_t - \mathcal{L}_{t+1}]$ is large, while at late training stages, the optimization reaches a plateau: $\mathbb{E}[\mathcal{L}_t - \mathcal{L}_{t+1}]$ decays, in contrast $\Delta_\sigma$ is constant and becomes relatively large, the best sparsification rate $r^*$ thus should be increasing during training.

## H  Scaling rule for hyperparameter tuning

First order momentum is commonly adopted in the optimization with DP-SGD, because momentum can alleviate oscillation and accelerate gradient descent (Sutskever et al., 2013), it is therefore believed to reduce the number of iterations of training and therefore achieve less privacy loss. However, for privacy-preserving training, momentum will also exaggerate the added Gaussian noise by incorporating current and all historical noise.

Denote velocity update: $v_{t+1} = \mu \cdot v_t + a \cdot \tilde{g}_{t+1}$, where $v, \mu, \tilde{g}$ denote perturbed velocity, momentum and perturbed gradients, respectively, common implementation of SGD with momentum includes $a = 1$, e.g. for Pytorch (Paszke et al., 2019a) or $a = -\gamma$, e.g. for Tensorflow (Abadi et al., 2016a). Using the expression of one step noise in Equation (4) and denoting by $\hat{v}_t$ the velocity after separating the noise, we have:

$$v_{t+1} - \hat{v}_{t+1} = (1 + \mu + \mu^2 + ... + \mu^t) \cdot a \mathcal{N}(0, C^2 \sigma^2 \boldsymbol{I}_d). \tag{117}$$

After many iterations, the scalar approximates a geometric series, i.e.:

$$v_{t+1} - \hat{v}_{t+1} \approx \frac{1}{1 - \mu} \cdot a \mathcal{N}(0, C^2 \sigma^2 \boldsymbol{I}_d), \tag{118}$$

with a residual decaying exponentially. Pulling the clipping bound $C$ out and forming the noise as $C(1 - \mu) \cdot a \mathcal{N}(0, \sigma^2 \boldsymbol{I}_d)$, we present a scaling rule for adjusting $C$ and $\mu$, namely the local optimal values of $C, \mu$ satisfy the same $C/(1 - \mu)$: the same amount of noise. As shown in Figure 10, the local optima of every column or row is located on the diagonal, where $C/(1 - \mu) = C^*/(1 - \mu^*)$ ($C^*, \mu^*$ are given in previous works).

The scaling rule suggests a workload saving pipeline for tuning, as it decomposes a 2D search into a line search: i) Set either $C$ or $\mu$ fixed and search for the other (search in a row or column); ii) Adjust $C, \mu$ jointly according to the scaling rule to find the optimal combination (searching along the diagonal).

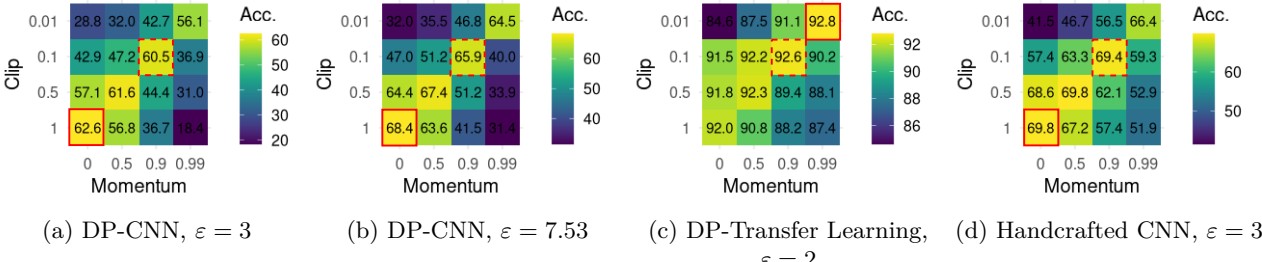

(a) DP-CNN, $\varepsilon = 3$  (b) DP-CNN, $\varepsilon = 7.53$  (c) DP-Transfer Learning, $\varepsilon = 2$  (d) Handcrafted CNN, $\varepsilon = 3$

Figure 10: Accuracy (%) under different combinations of clipping bound $C$ and momentum $\mu$. The diagonals represent combinations adjusted by the scaling rule based on values of $C$ and $\mu$ given in previous works. The red dashed square marks the given hyperparameters, and the red solid square the optimal hyperparameters.

We note that the scaling rule assumes noise being directly added to velocity, for projected DP-SGD where the noise has been projected, this rule may not be applied. Additionally, for some adaptive optimizers, e.g. Adam (Kingma & Ba, 2014), computing of velocity is implemented as follows:

$$v_{t+1} = \mu \cdot v_t + (1 - \mu)\tilde{g}_t, \tag{119}$$

$$v_{t+1}^* = v_{t+1}/(1 - \mu^t), \tag{120}$$

If we ignore the scaling in Equation (120) where the factor $1 - \mu^t$ increases rapidly to 1, we have:

$$v_{t+1} - \hat{v}_{t+1} = (1 + \mu + \mu^2 + ... + \mu^t) \cdot (1 - \mu)\mathcal{N}(0, C^2 \cdot \sigma^2 \boldsymbol{I}_d) \tag{121}$$

$$\approx \frac{1}{1 - \mu} \cdot (1 - \mu)\mathcal{N}(0, C^2 \cdot \sigma^2 \boldsymbol{I}_d) \tag{122}$$

$$= \mathcal{N}(0, C^2 \cdot \sigma^2 \boldsymbol{I}_d). \tag{123}$$

Here we see $C$ and $\mu$ are decoupled in term of the noise amount, which suggests that they can be tuned independently. So searching for the best combination can be decomposed into searching in a row followed by its column or vice versa, as the best $\mu$ is the same for different $C$ and vice versa, see Figure 11.

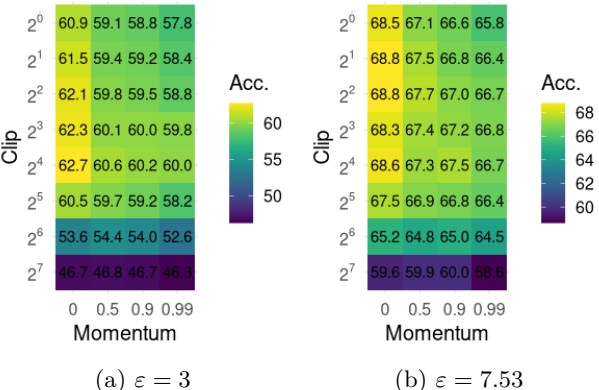

(a) $\varepsilon = 3$  (b) $\varepsilon = 7.53$

Figure 11: Accuracy (%) under different combinations of clipping bound and first order momentum. The network is DP-CNN and the optimizer is Adam (Kingma & Ba, 2014).

## I  DP-SGD with ranked sparsification

For ranked sparsification, we consider a straightforward implementation which ranks the (absolute) mean of the perturbed gradients of last epoch and sparsifies accordingly. As DP is robust to post-processing, such ranked sparsification does not consume the privacy budget.

Algorithm 4 describes the proposed ranked sparsification. Note that the noise added to the gradient mean estimation $z^e$ is not sparsified (Line 22), otherwise the coordinates that get masked out at the first iteration will receive gradient mean estimation as 0 and never get updated for all the remaining iterations, which will degrade the network. Adding noise to sparsified coordinates can give these coordinates a chance to be ranked in higher positions, while in turn the coordinates being updated but with low magnitude of their true gradient may get masked out in the next iteration.

However, it turns out that ranked sparsification *cannot* outperform random sparsification, more precisely it performs the same as RS. We further find that it seems the ranking is fundamentally random as the ranking is dominated by Gaussian noise. To demonstrate this, we run random sparsification and ranked sparsification for 100 epochs, then statistically analyze the distribution of how many times a parameter is masked out. The result shows the equivalency between these two strategies, which implies even averaging the perturbed gradients over a full epoch cannot sufficiently mitigate the noise added in gradient perturbation, see Figure 12.

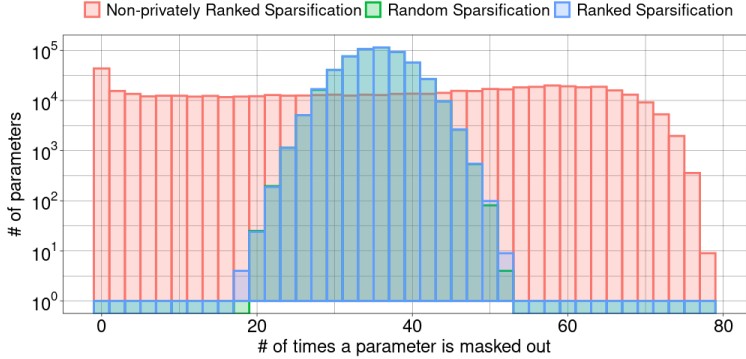

Figure 12: Histogram of the number of parameters vs. the number of times a parameter is masked out during 100 epochs training. For comparison we present non-privately ranked sparsification, i.e. ranking after excluding noise. Random sparsification and ranked sparsification mostly overlap.

This indicates that in the context of DP-SGD selecting the unimportant coordinates is difficult and stochasticity cannot be fully eliminated. Although there exist options to increase the precision of ranking, for example using public dataset (Zhou et al., 2021; Yu et al., 2021a) or via sparse vector techniques (SVT) and DP selection (Zhang et al., 2021; Dwork & Roth, 2014), there is probably a mismatch of the empirical distributions of public dataset and private dataset, while SVT and DP selection definitely contains randomness. However, as we show with RS, under such randomness, sparsification could still be beneficial for the optimization, which was not realized before.

## J  DP transfer learning with random sparsification

Differentially private (DP) transfer learning has recently gained popularity due to its effectiveness in various downstream tasks (Tramer & Boneh, 2021; Li et al., 2022; Yu et al., 2022; Bu et al., 2022). DP linear probing reuses the feature extractor, significantly reducing the gradient dimension (Abadi et al., 2016b; Tramer & Boneh, 2021). Li et al. (2022) demonstrated that DP fine-tuning of pre-trained large language models can achieve performance close to non-private fine-tuning. Meanwhile, Yu et al. (2022); Bu et al. (2022) showed that parameter-efficient fine-tuning outperforms full fine-tuning.

However, when applying RS to DP transfer learning, we have not observed significant performance improvement with high sparsification rates. The reason is that the configurations of DP transfer learning reduce the efficiency of RS. Based on Remark 3, RS is beneficial when the clipping bound $C$, noise multiplier $\sigma$, and gradient dimension $d$ are large, while the batch size $B$ is small. Although DP transfer learning adopts large models, linear probing or parameter-efficient methods constrain the gradient dimension $d$ to be less than 1% (Yu et al., 2022) or even 0.1% (Bu et al., 2022) of the original parameter space, rendering it small. Furthermore, since pre-trained networks converge quickly on downstream tasks, DP transfer learning methods typically employ hyperparameters such as large batch size $B$, small clipping bound $C$, and noise multiplier $\sigma$, making random sparsification less efficient. In contrast, in a training-from-scratch scenario, networks require more iterations to train. Large batch size $B$ and small noise multiplier $\sigma$ result in fewer iterations for a given privacy budget while small clipping bound $C$ limits the gradient magnitude, therefore they are not preferred in the hyperparameter tuning. And random sparsification is more favorable for training from scratch.

RS may be beneficial for DP full fine-tuning where the gradient dimension $d$ is sufficiently large. However, as this work focuses on the unique interaction between DP-SGD and RS, we leave a more comprehensive study of practical usage of RS for future research.

## K    Poisson sampling and random shuffle

In the analysis of DP-SGD Abadi et al. (2016b), Poisson sampling is used to induce the privacy amplification (Balle et al., 2018; Wang et al., 2019; Mironov et al., 2019). In the implementation of our baselines and many previous works (Papernot et al., 2021; Tramer & Boneh, 2021; Yu et al., 2021a), sampling is implemented by random shuffling the dataset and partitioning with fixed batch size, i.e. uniform sampling without replacement. It is reported that the model performance remains approximately the same under these two sampling schemes (Tramer & Boneh, 2021). In this work, we follow the baselines to conduct the experiments. To validate that the benefit of RS is consistent under this difference, we redo the experiments with DP-CNN on CIFAR10. The results are given in Table 3. We note that there is no significant difference in performance between "partition" and "Poisson" settings, and that the advantages of RS hold in both.

| Sampling | $\varepsilon$ | Baseline | RS (ours) | Difference |
|---|---|---|---|---|
| Parition | 3.0 | $62.8 \pm 0.10$ | $\mathbf{64.3 \pm 0.17}$ | +1.5 |
|  | 1.0 | $52.5 \pm 0.25$ | $\mathbf{55.1 \pm 0.11}$ | +2.6 |
| Poisson | 3.0 | $62.8 \pm 0.27$ | $\mathbf{64.6 \pm 0.25}$ | +1.8 |
|  | 1.0 | $52.8 \pm 0.24$ | $\mathbf{55.2 \pm 0.25}$ | +2.4 |

Table 3: Test accuracy (% ± SEM) before and after adopting random sparsification with different sampling schemes. Partition indicates random shuffling the dataset and partitioning into batches of fixed size per epoch.

