# OpenReview forum: "Improving Differentially Private SGD via Randomly Sparsified Gradients"
_TMLR — Accepted by TMLR_

### Review · Reviewer_gxEp · 2023-03-12

**Summary Of Contributions:**

The paper proposes a differentially private SGD algorithm with randomly sparsified (RS) gradients, i.e, randomly sample some coordinates of the gradient in each iteration and then use these selected coordinates to update the variables. The convergence rate of the proposed algorithm is established. In addition, an interesting phenomenon is found: the trade-off raised by RS in the context of DP-SGD is special, which manifests under two preconditions: gradient clipping and noisification.

**Audience:**

Yes

**Claims And Evidence:**

Yes

**Requested Changes:**

1. Please check the sampling scheme in Algorithm 1
2. Since the proposed algorithm is similar to random coordinate descent with DP, there need more discussions in the main paper to emphasize the difference between them.
3. In Remarks 1 and 2, it’s better to compare the established rates with the desirable rate.
4. Please go through the paper, some statements are not rigorous (e.g, Thm 3.4, “Combine Lemma 3.3, Lemma 3.4, and Equations 9, 10……”). Also, there are many typos in the paper. Pay attention to citing the equations (e..g, it should be “equation (7)” instead of “equation 7” in page 5).
5. $f(X)$ in equation (2) should be $\frac{1}{B}\sum_{x}g(x)$? Since the average gradient is used in Alg 1.


**Strengths And Weaknesses:**

Strength:
1. A new DP-SGD algorithm based on randomly sparsified gradients is proposed.
2. Convergence analysis for the proposed algorithm for the nonconvex case is provided. The convergence rate implies that DP-SGD with RS can recover the vanilla DP-SGD when $r=0$ and outperform the vanilla DP-SGD when $r>0$.
3. An interesting phenomenon is observed: Applying RS technique for noise SGD (without clipping) and SGD with clipping (without noising) will not improve the convergence rate.
The trade-off raised by RS in the context of DP-SGD is special.

Weakness
1. The basic DP-SGD algorithm used in the paper is proposed by Abadi et al.(2016). However, the sampling scheme used in Abadi et al.(2016) is Poisson sampling instead of uniform sampling without replacement, i.e., in each iteration $t$, we do a Bernoulli experiment with probability $q=L/N$ for each example $x_i$. With this sampling scheme, we cannot guarantee a minibatch with size $B$ is sampled from the whole dataset as we do in Algorithm 1. This problem will not affect the results of the main Theorem 3.4, but it will affect the results of the experiments since the variance of the Gaussian noise (see Theorem C.1 in Appendix C) is different for the two sampling schemes. Please check the results of Abadi et al.(2016), and let me know if I misunderstood.

2. The organization of the paper is not good enough, for example, Theorem 1.1 is not the main result of the paper, so it’s better to move it to Sec 3. Further, I would suggest the author first present the pseudo-code of DP-SGD with RS at the beginning of Sec 3 and then provide its convergence analysis, which is the main result of the paper. Then presenting the results in Sec 3.1 and 3.2.

---

> ### Author Response · Authors · 2023-04-06
> **Question by Authors**
>
> Dear Reviewer gxEp,
>
> Thank you for your valuable input.
>
> Before the rebuttal period begins, we would like to clarify a few points regarding your review to better prepare for our upcoming rebuttal. In the third point of your requested changes, you mentioned:
>
> > *In Remarks 1 and 2, it’s better to compare the established rates with the desirable rate.*
>
> In Remarks 1 and 2, we emphasize that our analysis in the corresponding cases demonstrates that random sparsification hinders convergence at any rate. This observation indicates that random sparsification is not a desirable approach for certain commonly used SGD schemes, rather than being specific to DP-SGD. We are unsure if this response addresses your concern. If not, could you please provide more information on what you mean by "desirable rate" and "established rate"?

---

> > ### Author Response · Authors · 2023-04-24
> > **Response by authors**
> >
> > We would like to thank Reviewer gxEp for the insightful comments and constructive feedback on our paper, and appreciate the time and effort the reviewer has dedicated to evaluating this work.
> >
> > We are grateful that Reviewer gxEp acknowledged the theoretical contribution of our work and thought our work interesting. Below, we address the specific concerns raised by the reviewer.
> >
> > > W1
> >
> > Abadi et al. (2016) employed Poisson sampling in their analysis to utilize the privacy amplification. However, in the implementation of many previous works including Abadi et al. (2016) and our baselines, dataset is randomly shuffled and partitioned into batches of a fixed size $B$, i.e. uniform sampling without replacement. There is an implicit assumption that the model performance will not be obviously affected. In the analysis, the accumulated Gaussian noise with multiple iterations of Poisson sampling will approximate the one with fixed batch size. In practice, model performance under these two scenarios are close as reported in Appendix D.4 in Tramer et al. (2021).
> >
> > In light of your comment, we change Algorithm 1 to Poission sampling and clarify in the experiment setup that we implement with fixed batch size by following previous works. In Appendix K, we add the experiment with DP-CNN on CIFAR-10 using Poisson sampling. The results confirm that the the benefit of RS is indeed consistent with our original findings.
> >
> > > W2
> >
> > As suggested, we have moved Theorem 1.1 to Section 3 and relabeled it as Theorem 3.1, to better align with the flow of the paper.
> > However, we would like to respectfully provide some clarifications regarding the organization of the remaining sections:
> >
> > 1. Moving the pseudo-code of DP-SGD with RS to the beginning of Section 3 is challenging as it involves gradual cooling and per-epoch randomization, which are only introduced in Section 4. Presenting the pseudo-code before these concepts are properly introduced could potentially create confusion.
> >
> > 2. The convergence analysis of RS is based on the results in Sections 3.1 and 3.2. Consequently, moving the convergence analysis before these sections would disrupt the logical flow and coherence of our presentation.
> >
> > Given these considerations, we have decided to maintain the current structure of the Sec 3.3 and the pseudo-code of DP-SGD with RS, as we believe it provides the most coherent and comprehensible presentation of our work.
> >
> > > RC1
> >
> > Please refer to W1.
> >
> > > RC2
> >
> > We have further clarified these distinctions in our updated discussion on random coordinate descent (Section 2), emphasizing our work's unique contributions. In the revised discussion, we highlighted that while Damaskinos et al. (2021) and Mangold et al.  (2022) also involve randomness in their DP coordinate descent studies, there are notable differences. Damaskinos et al.  (2021) focused on a generalized linear model, investigating the dual formulation of a convex problem, whereas Mangold et al.  (2022) studied convex optimization problems with precise coordinate-wise regularity measures for objective functions. Both works derive a modified update step (beside sparsification).
> >
> > In contrast, our work explores DP-SGD with RS in a general non-convex setting. We do not assume detailed characterization of the optimization landscape, e.g. coordinate-wise smoothness. And RS is directly applied on gradients computed with the loss function of the network's prediction. Our analysis is thereby more general and close to the practical application of DP-SGD.
> >
> > > RC3
> >
> > Please refer to our first comment block.
> >
> > >RC4
> >
> > Thank you for carefully reading our work. We have unified the equation citations and improved the descriptions of theorems for better readability, and have checked for remaining typos.
> >
> > > RC5
> >
> > In Algorithm 1 both gradient and noise are divided by the batch size $B$, we can also remove this coefficient and obtain Equation (2) and Equation (3). They are equivalent in terms of privacy.

---

> > > ### Comment · Reviewer_gxEp · 2023-05-08
> > > **Thank you for your response**
> > >
> > > Thank you for your reply. The response addresses my concerns. I do not have any more questions.

---

### Review · Reviewer_YD2n · 2023-03-24

**Summary Of Contributions:**

This work studies DP optimization with randomly sparsified (RS) gradient. The main contribution is theoretical and breaks the algorithm into two operations -- the per-sample gradient clipping and the noise addition. The authors claim that RS introduces a unique trade-off in the convergence bound and support their analysis with some empirical evidence.

**Audience:**

Yes

**Claims And Evidence:**

Yes

**Requested Changes:**

1. Rewrite Theorem 3.1 and all relevant parts.

2. Experiments on some fine-tuning tasks.

**Strengths And Weaknesses:**

Strengths: This work tackles an important and under-studied topic, i.e. the convergence of DP-SGD in the non-convex (deep-learning-related) setting. Therefore the impact can be significant. The overall structure is clear and convincing.

Weaknesses:

1. My major concern is the result in Theorem 3.1 and any theorems that use it. I believe **Theorem 3.1 is correct but not optimal nor sufficiently novel**. Similar analysis (under almost if not completely the same assumptions) has been presented in Appendix D of [1] and Theorem C.1 of [2]. In both cases, the learning rate is set to $1/G\sqrt{T}$. This is fundamentally different from Theorem 3.1: this work only shows $\|\nabla w\|^2\leq O(1/T)+constant$, which does not vanish as $T\to\infty$ so the algorithm does not converge; however, those work shows $O(1/\sqrt{T})$ after some trivial rewriting. Therefore, I believe Theorem 3.1 must be rewritten and further modification may be needed in following theorems and the trade-off. This won't address the novelty but improves the result.

2. Only toy datasets are experimented, where the improvement is rather marginal. This weakness does not necessarily mean a bad thing, but the authors should consider a different optimization regime **with pre-training**, given that all experiments here are training from random initialization. For example, CIFAR10/CIFAR100 can achieve 99%/90% accuracy under DP with 3 epochs fine-tuning. The computation cost is low and RS may have better results in this fine-tuning regime.

3. Related to the above fine-tuning, a popular topic is DP PEFT (parameter-efficient fine-tuning). This is briefly discussed in Section 2 but not extensively. For example, some particular methods like DP-LoRA, DP-Adapter, DP-BiTFiT should be discussed as alternative sparsified gradient. The comparision between DP PEFT and RS as examples: (1) RS is random and can invoke Lemma B.1. (2) RS cannot leverage the compute acceleration because the sparse mask is different at every iteration/epoch whereas DP PEFT sparsifies a fixed subset of parameters, so some tensor multiplication during back-propagation can be avoided. (3) RS and PEFT both enjoy communication benefit in distributed settings.

4. Minor: In the main text, try mentioning DP-Adam since RS directly works and some people don't regard Adam as SGD.



[1] Bu, Zhiqi, et al. "Automatic clipping: Differentially private deep learning made easier and stronger." arXiv preprint arXiv:2206.07136 (2022).

[2] Bernstein, Jeremy, et al. "signSGD: Compressed optimisation for non-convex problems." International Conference on Machine Learning. PMLR, 2018.

---

> ### Author Response · Authors · 2023-04-24
> **Response by authors**
>
> We would like to thank Reviewer YD2n for the insightful comments and constructive feedback on our paper, and appreciate the time and effort the reviewer has dedicated to evaluating this work.
>
> We are grateful that Reviewer YD2n recognizes the potential impact of our study and found our work clear and convincing. Below, we address the specific concerns raised by the reviewer.
>
> > W1
>
> We would like to note that in the context of DP-SGD, $T$ cannot go to infinity, as this would indicate pure noise in the gradient. However, we agree that a convergence rate of $O(1/\sqrt{T})$ should be better than $O(1/T) + constant$, considering the significant noise present in DP-SGD. We have updated Theorem 3.2 (Theorem 3.1 in the first submission) accordingly, and our conclusion remains unchanged.
>
> For Theorem 3.2 we need to specify the learning rate. Otherwise, the direction of inequality in Equation (31) may not be maintained. For other analyses, due to the gradient clipping, Equation (43) can be applied. Thus, we can express the learning rate explicitly in Theorems 3.4 and 3.5, and these analyses are not influenced by this suggestion.  We hope your major concern has been addressed.
>
> Ragarding the statement "Theorem 3.1 is correct but not... sufficiently novel," we would like to remind the reviewer about the acceptance criteria for TMLR outlined here (https://jmlr.org/tmlr/acceptance-criteria.html), specifically:
>
> *"...Nor should it form the basis for rejecting work on a method considered not “novel enough”, as novelty of the studied method is not a necessary criteria for acceptance. We explicitly avoid these terms (“significant”, “impactful”, “novel”)..."*
>
> Although we feel there are many novel and interesting aspects of this work, we do not need to rebut this here as it is not a review criterion.
>
> > W2
>
> DP with a pre-trained model has gained popularity recently.
> We have conducted some experiments in these scenarios, but do not find that a high sparsification rate significantly improves the performance (results from DP linear probing following [1] are given in Figure 4c). The reason is that DP linear probing as well as other parameter-efficient methods largely reduce the weight dimension $d$ for DP learning, which weakens the efficiency of random sparsification as discussed in Remark 3. Additionally, many works in this direction adopt large batch sizes $B$ and/or small clipping bound $C$, and/or small noise multiplier (i.e. fewer running iterations as also claimed by the reviewer), making random sparsification less efficient. We add a further discussion of this in Appendix J. We also include this as a limitation in Section 7.
>
> Random sparsification could potentially be beneficial to DP full fine-tuning with sufficiently large models due to very large $d$, but we have not yet observed this behavior.
>
> We would like to point out that transfer learning cannot entirely substitute training from scratch, especially in highly sensitive domains like medical data, where large-scale public datasets for pre-training are often unavailable. In these cases, training from scratch is essential and the method we propose can be useful. Additionally, a primary contribution of our work is identifying a special interaction between random sparsification and DP-SGD which is not observed by other SGD schemes. We believe this intriguing observation has significant implications for the broader research community and can inspire further study of DP-SGD.
>
> > W3
>
> Thanks for providing these interesting works. We have included a discussion of them in the revised related work section.
>
> > W4
>
> We have mentioned this in the implementation of RS, i.e. Section 4.
>
> ---------
> [1] Florian Tramer and Dan Boneh. Differentially private learning needs better features (or much more data). 2021.

---

### Review · Reviewer_6o4t · 2023-04-10

**Summary Of Contributions:**

This paper shows the positive impact that randomly sparsified gradients may have on the upper-convergence bounds of Differentially Private SGD. An improvement in the convergence rate can have a significant impact since the differential privacy requirements require increasingly larger batch-sizes that depend on the final number of iterations. Moreoever, sparsified gradients have some practical, appealing properties, because they are compressed versions of the original stochastic gradients.

More precisely, the authors show that, 1) spacification does not hurt the differential privacy guarantees (Thm 1.1) 2) spacification of the gradients degrades the upper-convergence bound of SGD for either noisz or clipped gradient but 3) improves the convergence upper bound of DP-SGD (SGD with clipped noisy gradient), specifically in the case of large noise. Finally, the authors validate their findings with experiments.

**Audience:**

Yes

**Broader Impact Concerns:**

This work does not describe any potential negative social impact of the results presented. However, since this paper is theoretical it may be interesting to wonder how these theoretical results may be misinterpreted to justify some actions or policies. For instance, the authors could develop the limitation of the differential privacy framework and to what extent it corresponds to a specific notion of privacy.

**Claims And Evidence:**

Yes

**Requested Changes:**

I do not have any significant requested change.


Minor notes:
- Caption of Figure 1: some quantities are only defined later  (e.g., $\xi_\sigma'$)
- Caption of Figure 3: with gradient clipping ~~and~~ or noisy SGD.


**Strengths And Weaknesses:**

Strengths:
- The message is clear and simple.
- The contribution is clear too.
- Improving theoretical upper-bounds (even by a constant) is very important in DP-SGD because of the DP guarantees limit the number of steps that can be performed (or ask for larger batch size).
- Experiments support the theory.

Weaknesses:
- The arguments are based on the fact that space gradients hurt the upper-convergence bound for clipped or noisy gradients. However, with another proof technique, one may be able to show that sparse gradients improve the convergence rates in that case. The authors could try to provide lower bounds to strengthen the significance of their claims and contributions.



**Question:**
Is there a way to improve Theorem 1.1? It seems the algorithm is seeing less information regarding the gradient and, thus, the data.
What is state of the art in terms of DP on CIFAR? It seems that your technique could be applied to SOTA techniques. If not, what is the issue?

---

> ### Author Response · Authors · 2023-04-24
> **Response by authors**
>
> We would like to thank Reviewer 6o4t for the insightful comments and constructive feedback on our paper, and appreciate the time and effort the reviewer has dedicated to evaluating this work.
>
> We are grateful that Reviewer 6o4t acknowledged our theoretical improvement over DP-SGD and found that our contribution is clear. Below, we address the specific concerns raised by the reviewer.
>
> >W1
>
> In response to this concern, we would like to clarify that our study focuses on a non-convex problem, which inherently makes it challenging to provide a lower bound for the convergence rate. In non-convex optimization problems, the landscape is often more complicated, and obtaining a lower bound would require additional assumptions.
>
> Considering the challenges associated with providing lower bounds for non-convex problems and the empirical evidence that supports our claims, we believe that our current analysis and results are valuable contributions to the field. Nonetheless, we appreciate your suggestion, in the discussion (Section 7), we add explicitly that we will continue to explore the possibility of obtaining lower bounds in future work.
>
> >Q1
>
> In principle, DP bounds the worst case, so Theorem 1.1 (now Theorem 3.1 in the revision) cannot be improved. Although random sparsification removes gradients from some coordinates, in the worst-case privacy scenario, original gradients at sparsified coordinates are close to zero, maintaining the original information during sparsification. Generally, we cannot assume a specific gradient distribution, making it impossible to derive smaller DP cost in the context of random sparsification.
>
> > Q2
>
> Our framework is implemented using models from Tramer and Boneh (2021) and Papernot et al. (2021).  These continue to be SOTA models: as reported in e.g. [1], the model of Tramer and Boneh (2021) dominates in the regime where $\varepsilon\leq 3$  without any assumption of additional data (e.g. pre-trained models).  We have updated our submission to include this information in Section 6.2.
>
> If the focus is solely on the DP learning accuracy on CIFAR allowing for additional prior information, applying fine-tuning or linear probing with pre-trained networks, i.e., DP transfer learning, is popular. We have conducted DP linear probing on CIFAR-10 in Figure 4c, but have not observed significant performance improvements with high sparsification rates. This is because linear probing largely reduces weight dimension $d$ for private learning, which weakens the efficiency of random sparsification as discussed in Remark 3. We add a detailed discussion of DP transfer learning with RS in Appendix J.
>
> > Minor Notes
>
> Thank you for carefully reading our work, we have improved these in the revision by clarifying the notations in the caption and change "and" to "or".
>
> > Broader Impact Concerns
>
> We appreciate the reviewer's concern about potential negative social impacts and the importance of discussing the limitations of differential privacy. In response, we have updated the discussion (Section 7) to emphasize that DP only reduces, not eliminates, the statistical dependency between its input and output. We also discuss the need to consider alternative privacy-enhancing techniques, such as cryptographic methods, depending on data subjects' specific requirements and expectations. This comprehensive view of the privacy landscape aims to prevent potential misinterpretations and promote responsible use of our results.
>
> ------------------
> [1] Soham De, Leonard Berrada, Jamie Hayes, Samuel L Smith, and Borja Balle. Unlocking high-accuracy differentially private image classification through scale. 2022.

---

### Author Response · Authors · 2023-04-24
**Overall response**

We have uploaded the revision based on the feedback and suggestions. The modifications are marked in red.

We would like to thank all reviewers for their valuable input and time.

---

### Decision · Action_Editors · 2023-06-13

**Recommendation:** Accept as is

**Comment:**

Based on the feedback from the reviewers, it appears that two of them have recommended accepting the paper after the rebuttal. While one reviewer still has a concern regarding algorithmic improvement, they acknowledge the interesting theoretical benefit shown by the Random Sparsification (RS) approach. Considering that the primary contribution of the work is mainly theoretical and has identified a unique and interesting interaction between random sparsification and DP-SGD, which sets it apart from other SGD schemes, I concur with the majority of the reviewers and recommend accepting the paper.

**Audience:**

Yes

**Claims And Evidence:**

Yes